# Hierarchical sequence-affinity landscapes shape the evolution of breadth in an anti-influenza receptor binding site antibody

Angela M Phillips[1,2]*[†], Daniel P Maurer[3,4]†, Caelan Brooks[5], Thomas Dupic[1], Aaron G Schmidt[3,4], Michael M Desai[1,5,6,7]*

[1]Department of Organismic and Evolutionary Biology, Harvard University, Cambridge, United States; [2]Department of Microbiology and Immunology, University of California, San Francisco, San Francisco, United States; [3]Ragon Institute of MGH, MIT, and Harvard, Cambridge, United States; [4]Department of Microbiology, Harvard Medical School, Boston, United States; [5]Department of Physics, Harvard University, Cambridge, United States; [6]NSF-Simons Center for Mathematical and Statistical Analysis of Biology, Harvard University, Cambridge, United States; [7]Quantitative Biology Initiative, Harvard University, Cambridge, United States

**\*For correspondence:**
angela.phillips@ucsf.edu (AMP);
mdesai@oeb.harvard.edu (MMD)

[†]These authors contributed equally to this work

**Abstract** Broadly neutralizing antibodies (bnAbs) that neutralize diverse variants of a particular virus are of considerable therapeutic interest. Recent advances have enabled us to isolate and engineer these antibodies as therapeutics, but eliciting them through vaccination remains challenging, in part due to our limited understanding of how antibodies evolve breadth. Here, we analyze the landscape by which an anti-influenza receptor binding site (RBS) bnAb, CH65, evolved broad affinity to diverse H1 influenza strains. We do this by generating an antibody library of all possible evolutionary intermediates between the unmutated common ancestor (UCA) and the affinity-matured CH65 antibody and measure the affinity of each intermediate to three distinct H1 antigens. We find that affinity to each antigen requires a specific set of mutations – distributed across the variable light and heavy chains – that interact non-additively (i.e., epistatically). These sets of mutations form a hierarchical pattern across the antigens, with increasingly divergent antigens requiring additional epistatic mutations beyond those required to bind less divergent antigens. We investigate the underlying biochemical and structural basis for these hierarchical sets of epistatic mutations and find that epistasis between heavy chain mutations and a mutation in the light chain at the $V_H$-$V_L$ interface is essential for binding a divergent H1. Collectively, this is the first work to comprehensively characterize epistasis between heavy and light chain mutations and shows that such interactions are both strong and widespread. Together with our previous study analyzing a different class of anti-influenza antibodies, our results implicate epistasis as a general feature of antibody sequence-affinity landscapes that can potentiate and constrain the evolution of breadth.

## Editor's evaluation

In this valuable study, the authors convincingly show that epistasis between mutations plays a role in the evolution of broadly neutralizing influenza antibodies. The authors provided accurate descriptions within the text and included a graphic summary focusing on the epistatic and non-epistatic models.

## Introduction

The diversity of influenza poses an ongoing public health challenge as vaccination and natural infection typically elicit immune responses that are highly strain-specific, and hence quickly lose efficacy as the virus evolves (*Kubo and Miyauchi, 2020*; *Angeletti and Yewdell, 2018*; *Neher and Bedford, 2015*; *Dugan et al., 2020*). This limited efficacy has garnered substantial interest in vaccination strategies that elicit broadly neutralizing antibodies (bnAbs) that neutralize diverse strains of influenza (*Corti et al., 2017*; *Angeletti and Yewdell, 2018*). Over the past two decades, there has been considerable effort to isolate and characterize anti-influenza bnAbs (*Whittle et al., 2011*; *Throsby et al., 2008*; *Corti et al., 2011*; *Dreyfus et al., 2012*). These bnAbs target various conserved epitopes on the hemagglutinin (HA) influenza surface glycoprotein, including the receptor binding site (RBS) (*Whittle et al., 2011*), the stem or stalk domain (*Corti et al., 2011*; *Dreyfus et al., 2012*; *Ekiert et al., 2009*), the lateral patch (*Raymond et al., 2018*), and the membrane-proximal anchor site (*Guthmiller et al., 2022*). BnAbs also vary in germline gene usage and breadth, with some binding several strains within an HA subtype and others binding nearly all characterized influenza strains (*Corti et al., 2017*).

Despite the immense body of work on influenza bnAbs, we still do not fully understand the evolutionary processes through which they mature (*Jiang et al., 2013*; *Horns et al., 2019*; *Sangesland and Lingwood, 2021*). Our strategies to elicit them therefore remain limited. It is clear, however, that distinct antibodies with diverse sequences can target the same HA epitope and evolve broad reactivity (*Schmidt et al., 2013*; *Dreyfus et al., 2012*; *Ekiert et al., 2009*; *Schmidt et al., 2015a*; *Wu et al., 2020a*). This redundancy suggests that there are many possible evolutionary pathways to influenza bnAbs. Still, the relatively low frequencies at which they are observed in human repertoires following vaccination suggest that there are factors constraining their maturation that we do not yet fully appreciate (*Wu et al., 2017*; *Bajic et al., 2019*; *Horns et al., 2020*; *Abbott et al., 2018*; *Andrews et al., 2015*).

High-throughput mutagenesis approaches are widely used as a tool to understand key properties shaping the evolution of numerous proteins (*Starr et al., 2017*; *Miton and Tokuriki, 2016*; *Bank et al., 2015*; *Phillips et al., 2018*). This work has revealed that new mutations can differentially impact distinct protein functions and often interact non-additively (i.e., *epistatically*), potentially constraining the order in which they can occur (*Weinreich et al., 2006*; *Gong et al., 2013*; *Sailer and Harms, 2017*). For antibodies, high-throughput mutagenesis studies have largely been limited to examining the effects of single mutations, either through saturating mutagenesis (e.g., deep mutational scanning) of relatively small regions (*Wu et al., 2017*; *Adams et al., 2016*; *Forsyth et al., 2013*) or through random mutagenesis (e.g., error-prone PCR) (*Li et al., 2018*; *Amon et al., 2020*; *Bowers et al., 2018*). These methods examine the local mutational landscape of a particular antibody, or in other words, how single mutations can change affinity or breadth. The advantage of these methods is that the sequences analyzed are relatively unbiased, particularly for saturating mutagenesis, and thus one can surmise why particular mutations occurred naturally. For example, this approach identified many single amino acid substitutions in the anti-influenza bnAb C05 that improve affinity to different subsets of strains but typically reduce breadth (*Wu et al., 2017*).

A key limitation of saturating mutagenesis approaches is that they cannot probe how epistatic interactions between mutations might constrain antibody evolutionary trajectories, which typically involve multiple mutations (*Victora and Nussenzweig, 2012*). Because antibodies acquire numerous mutations and experience fluctuating selection pressures on short timescales (*Victora and Nussenzweig, 2012*; *Smith et al., 2004*), they are necessarily distinct from other proteins for which epistasis has been studied. Moreover, they bind antigens through disordered loops, in contrast to the structured active sites of most enzymes, and they are relatively tolerant to mutations (*Braden et al., 1998*; *Burks et al., 1997*; *Chen et al., 1999*; *Klein et al., 2013*; *Corti and Lanzavecchia, 2013*). Further, the evolutionary dynamics of affinity maturation are defined by discrete rounds of mutation and selection compared to the more continuous processes most proteins are subject to, and thus mutations that occur concurrently are selected based on their collective rather than individual effects (*Victora and Nussenzweig, 2012*; *Unniraman and Schatz, 2007*). For these reasons, the evolutionary constraints on antibodies may be unique.

The few studies that have examined epistasis in antibodies indicate that it is a key determinant of affinity (*Schmidt et al., 2013*; *Phillips et al., 2021*; *Pappas et al., 2014*; *Adams et al., 2019*). For example, multiple studies have identified mutations that interact synergistically to bind an antigen

(*Schmidt et al., 2013*; *Phillips et al., 2021*; *Pappas et al., 2014*). Still, most of this work has focused on interactions between a small subset of mutations (*Schmidt et al., 2013*; *Xu et al., 2015*). Addressing the prevalence and general importance of epistasis in shaping antibody evolution will require more comprehensive combinatorial mutagenesis strategies that sample combinations of mutations present in each somatic antibody sequence (*Phillips et al., 2021*). These combinatorial strategies, however, do not capture epistasis with other mutations that could have occurred in alternative evolutionary pathways, which will require integrating combinatorial mutagenesis with the saturating mutagenesis methods described above.

In previous work, we systematically mapped the relationship between antibody sequence and affinity (the sequence-affinity landscape) across mutational landscapes relevant for the somatic evolution of two stem-targeting bnAbs of varying breadth, CR6261 and CR9114 (*Phillips et al., 2021*). We found that affinity was determined by nonadditive interactions between mutations, and that such epistasis could both constrain and potentiate the acquisition of breadth. Notably, the nature of this epistasis varied considerably between the two bnAbs. For CR6261, epistatic interactions were similar for binding distinct group 1 strains, thus evolutionary pathways could simultaneously improve in affinity to divergent antigens. For CR9114, increasingly divergent antigens required additional epistatically interacting mutations such that evolutionary pathways were constrained to improve in affinity to one antigen at a time. The distinct topologies of these sequence-affinity landscapes result from differences between the various antigens and the mutations that are required for binding.

Although anti-stem bnAbs are among the broadest influenza bnAbs characterized, they are a small and biased subset of the influenza antibody response. Despite the presence of anti-stem antibodies in human sera (*Yassine et al., 2018*) and the ability to drive viral escape mutants in vitro (*Wu et al., 2020b*), the stem is minimally mutated amongst circulating viral strains and does not appear to be evolving in response to these antibodies, possibly due to the high concentration of antibody required for protection (*Ellebedy, 2018*; *Han et al., 2021*) and immune pressure. In contrast, due to the small size of the RBS pocket, RBS-directed bnAbs frequently make contacts with immunodominant epitopes surrounding the pocket that have substantial antigenic variation (*Whittle et al., 2011*; *Schmidt et al., 2015b*; *Lee et al., 2014*; *Krause et al., 2011*; *Guthmiller et al., 2021*). Although RBS-directed bnAbs have a relatively narrower reactivity profile compared to anti-stem bnAbs, they are potently neutralizing, do not require effector functions for potent in vivo protection as do anti-stem bnAbs (*Corti et al., 2011*; *DiLillo et al., 2014*), and have evolved broad recognition despite the accumulation of antibody escape mutations in the periphery of the RBS. Further, RBS-directed bnAbs can mature from diverse germline $V_H$ and $V_L$ genes (*Schmidt et al., 2015b*), suggesting that there are likely numerous evolutionary pathways to target this epitope. Thus, to understand more generally how epistasis constrains bnAb evolution, here we consider RBS-directed bnAbs as they target an entirely different epitope under distinct immune selection pressures.

Specifically, we examine the influence of epistasis on the evolution of a well-characterized RBS-directed bnAb, CH65, which binds and neutralizes diverse H1 strains (*Whittle et al., 2011*; *Schmidt et al., 2013*). CH65 was isolated from a donor 7 days post-vaccination. The unmutated common ancestor (UCA) and an early intermediate (I-2) have moderate affinity for a subset of H1 strains that circulated early in the donor's lifetime (*Whittle et al., 2011*; *Schmidt et al., 2015a*). The affinity-matured CH65 has 18 mutations throughout the light ($V_L$) and heavy ($V_H$) variable regions, which improve affinity to strains that circulated early in the donor's life and confer affinity to antigenically drifted strains (*Schmidt et al., 2013*; *Xu et al., 2015*). Thus, CH65 evolved to acquire affinity to emerging strains without compromising affinity for previously circulating strains.

The structural changes upon affinity maturation of CH65 and clonally related antibodies have been extensively characterized (*Whittle et al., 2011*; *Schmidt et al., 2013*; *Schmidt et al., 2015a*). This work showed that CH65 primarily matures by preconfiguring the HCDR3 loop into its binding conformation, thereby minimizing the conformational entropic cost of binding (*Schmidt et al., 2013*). Importantly, none of the 18 somatic mutations are in the HCDR3; rather, key mutations in HCDR1, HCDR2, and LCDR1 result in contacts that stabilize the HCDR3 loop in its binding-compatible conformation (*Schmidt et al., 2013*; *Xu et al., 2015*). This structural characterization, in addition to molecular dynamics simulations, identified specific mutations that are critical for breadth, including some that interact synergistically to stabilize the HCDR3 loop (*Schmidt et al., 2013*; *Xu et al., 2015*). In theory, such epistasis could constrain bnAb evolution by requiring multiple mutations to confer a selective

advantage, or alternatively, it could compensate for the deleterious effects of other mutations and favor selection of bnAbs.

Given that many mutations in CH65 are important for imparting affinity to HA, including those at distant sites (*Schmidt et al., 2013*), we hypothesized that there are many sets of epistatic mutations in CH65 not previously identified, particularly because long-range epistatic interactions are difficult to predict from structural analyses alone. In contrast to the anti-stem bnAbs described above (*Dreyfus et al., 2012*; *Lingwood et al., 2012*), CH65 engages HA through both light and heavy chain contacts and requires mutations in both chains to bind divergent antigens (*Schmidt et al., 2013*; *Xu et al., 2015*). Thus, it is likely that mutations interact epistatically both within and between the heavy and light chains. Despite the potential importance of these interactions in shaping the evolution of CH65, and the numerous other antibodies that engage antigens using both chains (*Corti et al., 2011*; *Schmidt et al., 2015a*; *Xiao et al., 2019*; *Ekiert and Wilson, 2012*), characterizations of inter-chain epistasis have so far been limited to small sets of a few mutations (*Schmidt et al., 2013*; *Xu et al., 2015*). Although these smaller datasets have revealed some important inter-chain epistatic interactions, they measure a subset of interactions selected based on structural data, and thus we still do not know the magnitude or prevalence of this epistasis and hence how important it is in shaping antibody evolution.

Here, to elucidate the role of epistasis (both inter- and intra-chain) in shaping the evolution of an RBS antibody, we systematically characterize the CH65 sequence-affinity landscape. Specifically, we generate a combinatorially complete antibody library containing all possible evolutionary intermediates between the UCA and the mature somatic sequence (N=$2^{16}$ = 65,536) and measure affinity to three antigenically distinct H1 strains to assess how epistasis can shape evolutionary pathways, leading to varying levels of breadth. We find that strong high-order epistasis constrains maturation pathways to bind antigenically distinct antigens. Although fewer epistatic mutations are needed to bind an antigen similar to that bound by the UCA, these sets of mutations overlap with those required to bind a more divergent antigen. Collectively, these landscapes provide mechanistic insight into how affinity maturation responds to an evolving epitope and how exposure history can influence future immune responses. In combination with our previous work on anti-stem bnAbs (*Phillips et al., 2021*), this work shows how epistasis can differentially impact the evolutionary trajectories of bnAbs of varying breadth, epitope, and variable chain gene usage.

## Results

To comprehensively examine how epistasis may have shaped the evolution of CH65, we generated a combinatorially complete antibody library comprising all possible evolutionary intermediates from the UCA to CH65. This library contains all possible combinations of mutations present in both the variable heavy and light chains of CH65, less two mutations (Q1E and S75A in V$_H$) distant from the paratope that do not significantly impact binding affinity (*Figure 1A*, *Figure 1—figure supplement 1*) or physically interact with other residues. Removing these mutations results in a final library size of $2^{16}$, which is within the throughput limit of our methods.

To profile the breadth of the corresponding antibody library, we first transform this combinatorial plasmid library into yeast for antibody surface display in a single-chain variable fragment (scFv) format (*Boder and Wittrup, 1997*). We then use Tite-Seq (*Adams et al., 2016*), a high-throughput method that couples flow cytometry with sequencing, to measure equilibrium binding affinities to three H1 strains bound by CH65. We chose these strains to sample varying levels of antigenic change (*Smith et al., 2004*; *Bedford et al., 2014*): they include a strain that circulated early in the donor's lifetime (A/Massachusetts/1/1990, 'MA90') and a strain that circulated 16 years later (A/Solomon Islands/3/2006, 'SI06') (*Schmidt et al., 2013*). Additionally, because affinity maturation has been shown to confer binding to antigens that escape less mutated members of the same lineage (*Muecksch et al., 2021*), we drove viral escape of MA90 in vitro using the UCA and found that CH65 could bind to the resulting strain (A/Massachusetts/1/1990 G189E, 'MA90-G189E') that escapes the UCA. We use the MA90-G189E antigen to profile incremental antigenic change from MA90 (one direct escape mutation), whereas SI06 represents more substantial antigenic change during natural evolution, including loss of K133a and the mutation E156G in the RBS.

For each of these three antigens, we used Tite-Seq to measure equilibrium binding affinities for all $2^{16}$ variants in biological duplicates (*Figure 1—figure supplement 2*). We log-transform the

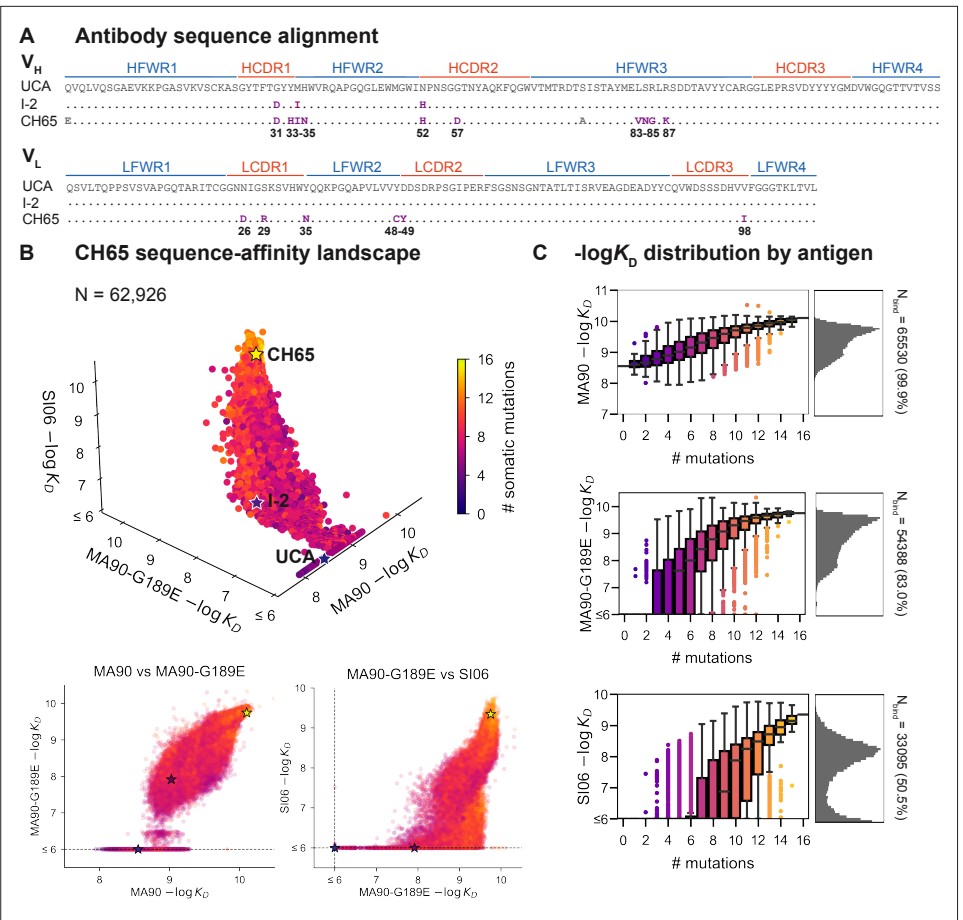

**Figure 1.** CH65 sequence-affinity landscape. (**A**) Alignment of unmutated common ancestor (UCA), I-2, and CH65 $V_H$ (top) and $V_L$ (bottom) sequences. Mutations of interest are shown in purple and are numbered; gray mutations do not impact affinity and were excluded from the library. (**B**) -log$K_D$ for ~$2^{16}$ variants to each of the three antigens. Each point represents the mean -log$K_D$ of biological duplicates and is colored by the number of somatic mutations in the corresponding variant. The UCA, I-2, and CH65 are annotated as stars; N = 62,926 after filtering poor $K_D$ measurements from the Tite-Seq data (see 'Materials and methods'). Two-dimensional representations of the data are shown below the three-dimensional plot. (**C**) Distribution of -log$K_D$ for each antigen. Left: variant -log$K_D$ grouped by the number of somatic mutations; Right: -log$K_D$ histograms for variants that bind each antigen, with total number of binding variants (N) indicated on plot.

The online version of this article includes the following source data and figure supplement(s) for figure 1:

**Source data 1.** CH65 library expression and -log$K_D$ to MA90, MA90-G189E, and SI06.

**Source data 2.** Isogenic flow cytometry measurements of -log$K_D$ and expression for select CH65 variants.

**Figure supplement 1.** CH65 mutation reversion.

**Figure supplement 2.** Tite-Seq workflow.

**Figure supplement 3.** Tite-Seq $K_D$ quality control and isogenic measurements.

**Figure supplement 4.** CH65 library expression.

binding affinities and report -log$K_D$, which is proportional to the free energy change of binding (and is thus expected to combine additively) (*Wells, 1990*; *Olson et al., 2014*). For each antigen, the Tite-Seq -log$K_D$ correlate well between biological replicates (r ~ 0.98 for all antigens; *Figure 1—figure supplement 3A*) and accurately reflect isogenic measurements made by flow cytometry (r = 0.97; *Figure 1—figure supplement 3B*), as well as recombinant IgG affinity measurements made by biolayer interferometry (r = 0.86; *Figure 1—figure supplement 3C*).

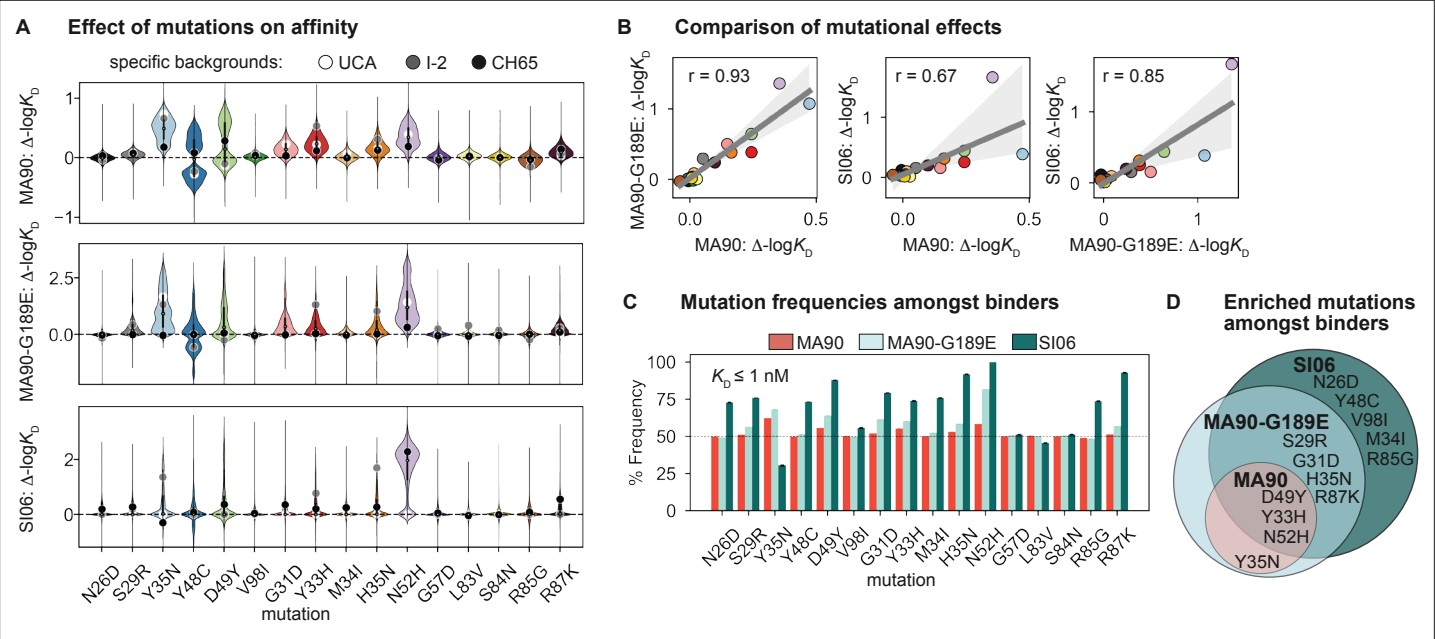

**Figure 2.** Mutational effects on affinity. (**A**) Change in -log$K_D$ resulting from each mutation on all ~$2^{15}$ genetic backgrounds. Impact of mutation on the unmutated common ancestor (UCA), I-2, and the CH65 genetic backgrounds are represented by white, gray, and black points, respectively. (**B**) Correlation of mean effect on -log$K_D$ for MA90, SI06, and MA90-G189E resulting from each mutation. Regression line and 95% confidence intervals are shown in gray. Mutations are colored as in (**A**). (**C**) Frequency of each mutation amongst variants that bind a given antigen with $K_D \leq 1$ nM. Error bars correspond to standard deviation across bootstrapped data (N = 10). (**D**) Mutations present at >55% frequency (p-value<0.05 from one-sided *t*-test) amongst binders for each antigen. *Figure 2—figure supplement 1.* Change in -log$K_D$ resulting from each mutation as a function of the number of other mutations present.

The online version of this article includes the following figure supplement(s) for figure 2:

**Figure supplement 1.** Change in -log$K_D$ resulting from each mutation as a function of the number of other mutations present.

## CH65 sequence-affinity landscape

Broadly, we find that increasingly divergent antigens require additional mutations to confer antigen binding. Consistent with previous work (*Schmidt et al., 2013*; *Xu et al., 2015*), the UCA has weak affinity for MA90 but does not bind MA90-G189E or SI06; I-2 (which contains G31D, M34I, and N52H (V$_H$)) has improved affinity to MA90, weak affinity to MA90-G189E, and does not bind SI06; and CH65 has near maximal affinity amongst library variants for all three antigens (*Figure 1B*). While the entire library binds MA90, ~83% of variants bind MA90-G189E and ~51% of variants bind SI06 (*Figure 1C*). For all antigens, affinity is higher for more mutated variants, except for a subset of highly mutated variants that do not bind SI06 (*Figure 1B*, bottom right). There are ~2000 variants that bind MA90 with reduced affinity relative to the UCA; none of these variants have detectable affinity for SI06, and only one has detectable affinity for MA90-G189E (*Figure 1B*). Further, all variants that bind SI06 also bind MA90-G189E (*Figure 1B*) as variants can bind MA90-G189E with fewer mutations than SI06 (*Figure 1C*). This 'hierarchical' or 'nested' pattern, where mutations that enable binding to more antigenically divergent strains are dependent on mutations that enable binding to less divergent strains, is reminiscent of what we observed previously for the anti-stem bnAb CR9114 (*Phillips et al., 2021*), despite the comparatively subtle differences between the antigens examined here (83–96% epitope identity versus 52–61% for the CR9114 antigens) (*Dreyfus et al., 2012*; *Schmidt et al., 2015a*).

## Mutational effects on CH65 affinity and breadth

To understand how specific mutations shape the sequence-affinity landscape, we computed the change in affinity resulting from each of the 16 mutations on all $2^{15}$ genetic backgrounds at the other 15 sites. This analysis reveals that several mutations improve affinity to MA90 and MA90-G189E (e.g., Y35N, Y48C, D49Y (V$_L$) and G31D, Y33H, H35N, N52H (V$_H$)), and some of these distributions are multi-modal, indicating that their effect on affinity depends on the presence of other mutations (*Figure 2A*).

Consistent with this, some mutations improve affinity to MA90 and/or MA90-G189E on the UCA or I-2 backgrounds (e.g., Y35N (V$_L$) and Y33H, H35N (V$_H$)) and others on the CH65 background (Y48C, D49Y (V$_L$)). For SI06, N52H dramatically improves affinity and most variants lacking this mutation do not have detectable affinity. Thus, several mutations (e.g., Y35N (V$_L$)) improve affinity to SI06 in the I-2, but not the UCA, background (*Figure 2A*). In general, the effects of these mutations correlate between the different antigens, with mutations affecting affinity more substantially for MA90-G189E and SI06 compared to MA90 (*Figure 2B*).

To assess which mutations confer affinity to a particular antigen, we computed the frequency of each mutation amongst binding variants in the library. Consistent with the landscapes in *Figure 1B*, we observe that the mutations enriched amongst binders form a hierarchical pattern between the antigens (*Figure 2C*). For example, a few mutations are enriched (≥55% frequency) amongst variants with nanomolar affinity for MA90 (e.g., Y35N, D49Y (V$_L$) and Y33H, N52H (V$_H$)), a few additional mutations are enriched amongst MA90-G189E binders (e.g., S29R (V$_L$) and G31D, H35N, R87K (V$_H$)), and still additional mutations are enriched amongst SI06 binders (e.g., N26D, Y48C, V98I (V$_L$) and M34I, R85G (V$_H$)) (*Figure 2D*). Thus, except for Y35N (V$_L$), which is interestingly depleted amongst SI06 binders (*Figure 2—figure supplement 1*), the mutations that enhance affinity to the three antigens form a hierarchical pattern.

We next characterized how epistasis between these mutations might impact affinity and result in this hierarchical pattern of breadth. To this end, we fit our measured -log$K_D$ to a standard biochemical model of epistasis (*Sailer and Harms, 2017*), which is a linear model defined as the sum of single mutational effects and epistatic terms up to a specified order (see 'Materials and methods'). Using a cross-validation approach, we find that the optimal order model for affinity is fourth-order for MA90 and fifth-order for MA90-G189E and SI06, and we report coefficients at each order from these best-fitting models (*Figure 3A*). The magnitude and sign of these coefficients correspond to effects on -log$K_D$: for example, a second-order term of +1 means that two mutations occurring together improve -log$K_D$ by 1 unit, beyond the sum of their first-order effects. For all three antigens, we find widespread epistasis between mutations in the same chain and between mutations in different chains, with many epistatic terms exceeding first-order effects in magnitude (*Figure 3*). In contrast to our previous work on variable heavy-chain-only antibody landscapes (*Phillips et al., 2021*), we find many strong epistatic interactions between mutations that are too distant to physically interact (*Figure 3—figure supplements 1–4*).

## Structural and biophysical basis of epistasis in CH65

Because there are substantial long-range epistatic interactions, our combinatorial approach identifies numerous interactions not previously known, in addition to confirming the few interactions characterized in earlier work (e.g., Y48C and D49Y (V$_L$) have strong synergistic epistasis) (*Schmidt et al., 2013*; *Xu et al., 2015*). Here, we find strong epistasis between the I-2 mutations (G31D, M34I, N52H (V$_H$)), neighboring mutations (Y33H, H35N (V$_H$)), mutations known to stabilize light chain contacts (Y48C, D49Y) (*Schmidt et al., 2013*), as well as an uncharacterized light chain mutation (Y35N). When we examine the structural context of this epistasis, we find that mutations with strong first-order and epistatic effects often make contact with either HA or with the HCDR3 that engages the RBS (*Figure 3B*, *Figure 3—figure supplement 5*). This suggests that the effects of these mutations are either mediated through the contacts that they make with HA or through affecting the HCDR3 loop conformation. These mutations interact epistatically for each of the three antigens, though the magnitude of epistasis is higher for SI06 (explaining ~34% of the variance in $K_D$, relative to ~24% for MA90, and ~25% for MA90-G189E, see *Figure 3—figure supplement 6*).

Importantly, these epistatic interactions are essential for the acquisition of affinity to both MA90-G189E and SI06. To investigate the molecular details of this epistasis, we compared the previously determined crystal structures of the unbound UCA, I-2, CH65, and CH65 bound to SI06 (*Whittle et al., 2011*; *Schmidt et al., 2013*; *Lee et al., 2015*), as well as newly determined crystal structures of the unbound UCA with Y35N (V$_L$) and I-2 with Y35N (V$_L$) and H35N (V$_H$). Additionally, we produced several variants as recombinant IgG to assay the binding kinetics by biolayer interferometry. To mimic the Tite-Seq system, IgG was bound to biosensors and assayed for binding to full-length trimeric HA (*Figure 4—figure supplements 2–4*). Here, we focus on binding kinetics of minimally mutated variants that confer affinity to each antigen but binding of 12 variants was assayed to all three antigens at varying temperatures (*Figure 4—figure supplement 5*).

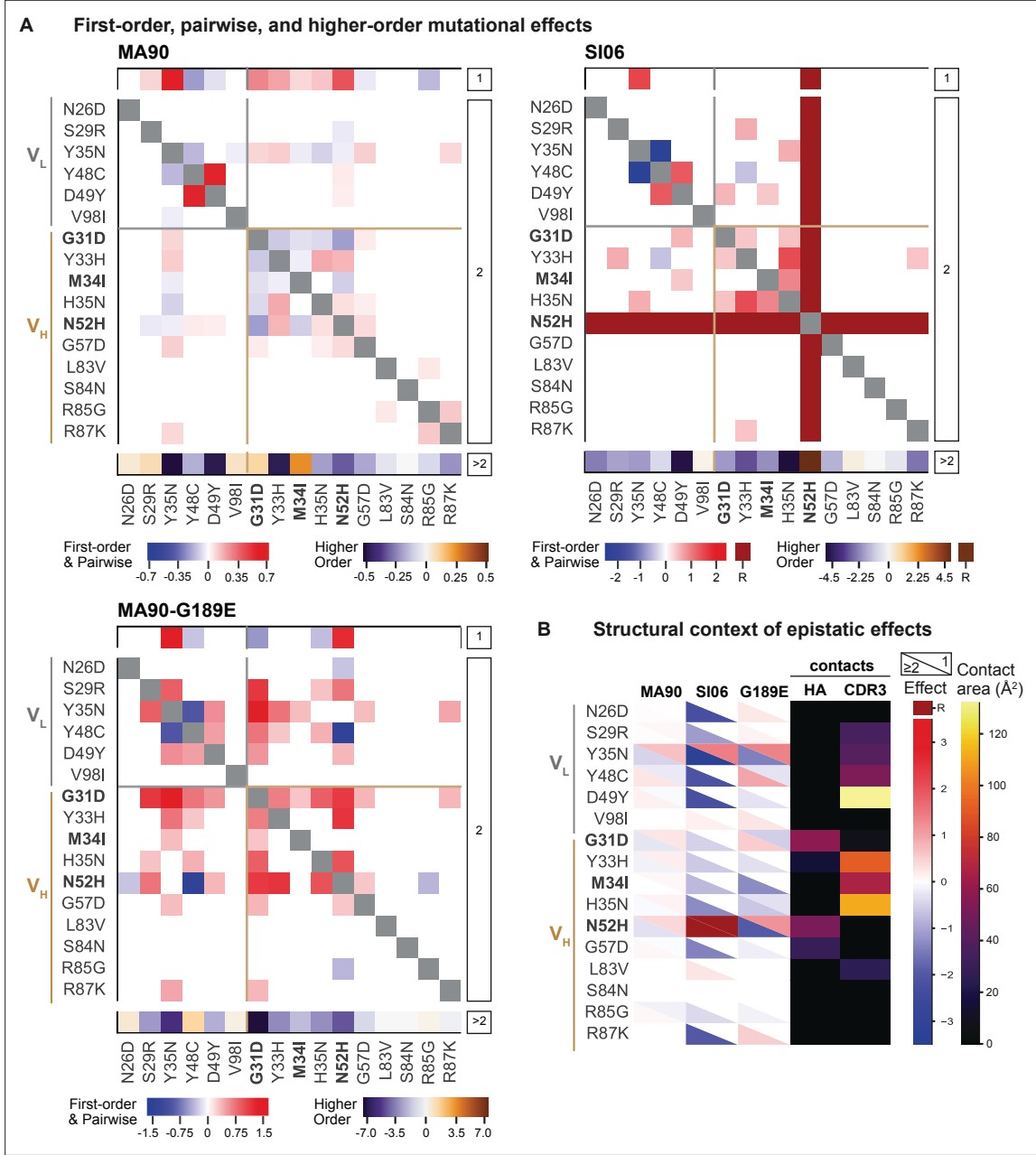

**Figure 3.** Epistatic coefficients for biochemical model of epistasis. (**A**) Significant first-order, pairwise, and higher-order mutational effects for each of the 16 mutations inferred from the optimal order model for each antigen. Higher-order effects are reported as a sum. Mutations present in I-2 are shown in bold. 'R' indicates that the mutation is required for binding (defined as being present in ≥90% of binding variants) and is thus excluded from the epistasis inference. (**B**) Structural context of significant first-order and epistatic effects. For each mutation, the upper triangle shows the first-order effect, the lower triangle shows the sum of the pairwise and higher-order effects, and the contact surface area with HA and HCDR3 are shown in the fourth and fifth columns. Significance in (**A**) and (**B**) indicates the coefficient 95% CIs do not include zero, see 'Materials and methods' and *Figure 3—source data 1*. *Figure 3—figure supplement 1*. Pairwise effects versus distance.

The online version of this article includes the following source data and figure supplement(s) for figure 3:

**Source data 1.** Interaction model coefficients for CH65.

**Figure supplement 1.** Pairwise effects versus distance.

**Figure supplement 2.** Biochemical epistasis within heavy and light chains and between chains.

**Figure supplement 3.** Epistatic coefficients for statistical model of epistasis.

**Figure supplement 4.** Comparison of coefficients in biochemical and statistical models.

*Figure 3 continued on next page*

*Figure 3 continued*

**Figure supplement 5.** First-order and epistatic effects plotted on co-crystal structure for each antigen.

**Figure supplement 6.** Variance partitioning of statistical epistasis coefficients by order of interaction.

The I-2 intermediate (which contains G31D, M34I, N52H ($V_H$)) is amongst the least-mutated variants that binds MA90-G189E (*Figure 1B*). The N52H mutation, which substantially improves affinity to MA90-G189E (*Figure 4A*), would potentially clash with one of the binding-incompatible HCDR3 conformations observed in crystal structures of the UCA and I-2; thus, this mutation may increase the occupancy of the binding-compatible HCDR3 conformation (*Figure 4—figure supplement 1A*). In the bound state, N52H π-stacks with Y33H and hydrogen bonds with G31D (*Figure 4B and C*, *Figure 4—figure supplement 1B*). Consequently, N52H and G31D, which together provide high affinity for MA90-G189E (*Figure 4A*), form a network of interactions between HCDR1, HCDR2, and the 150-loop of HA to stabilize the binding interaction (*Figure 4C*). Thus, while N52H alone confers affinity to MA90-G189E, G31D and M34I (I-2) reduce the dissociation rate by ~3.5–5-fold to improve affinity (*Figure 4C*, *Figure 4—figure supplement 5*). Notably, these interactions are distant (~15–20 Å) from the residue conferring viral escape, precluding any direct interaction (*Figure 4C*). Though these mutations are also important for affinity to the antigenically drifted SI06, they are insufficient to confer appreciable affinity in the absence of other epistatic mutations (*Figure 4D*, top).

In examining the minimally mutated variants that can bind SI06, we find that Y35N in the light chain and H35N in the heavy chain interact synergistically with the I-2 mutations (G31D, M34I, N52H) to confer affinity to SI06 (*Figure 4D*, bottom). Thus, the hierarchical sets of mutations that confer broad reactivity to these antigens (*Figure 2D*) do so through epistasis. In particular, the germline residue Y35 in the light chain framework (FWR) 2 is part of a cluster of aromatic residues at the $V_H$-$V_L$ interface and makes π-stacking, methionine–aromatic, and hydrogen bonding interactions between LFWR2 and LCDR3, HCDR3, and HFWR4 (*Figure 4E*). The somatic mutation Y35N effectively removes these interactions with the HCDR3. Although the loss of the aromatic moiety from tyrosine to asparagine likely has a destabilizing effect, we attribute the observed changes in affinity to the loss of hydrogen bonding between LFWR2 and HCDR3; this is in part because the lineage member CH67, which has similarly broad reactivity, acquires a Y35F mutation upon affinity maturation that is only a removal of a hydroxyl group, preventing the ability to form hydrogen bonding interactions through the side chain (*Schmidt et al., 2013*). Similarly, H35N removes a methionine–aromatic interaction, known to have a stabilizing effect in proteins (*Valley et al., 2012*), between the HFWR2 and HCDR3 (*Figure 4E*). Addition of Y35N and H35N into the UCA background did not confer affinity to SI06 (*Figure 4D and E*). However, the addition of H35N into the I-2 background produced weak but detectable affinity with an association rate that was improved upon addition of Y35N (*Figure 4D and E*, *Figure 4—figure supplement 6*). Notably, while Y35N confers affinity to SI06 for variants with few somatic mutations, the magnitude of this effect diminishes as the number of mutations increases. Indeed, Y35N is depleted amongst the highest affinity variants (*Figure 2C*, *Figure 2—figure supplement 1*) and in the context of a mutated background decreased the association rate and overall affinity (*Figure 4E*), suggesting that Y35N, which removes inter-chain contacts, is likely only beneficial during early rounds of affinity maturation.

Previous studies on this lineage (*Schmidt et al., 2013*; *Xu et al., 2015*) showed that HCDR3 rigidification contributed to high-affinity binding to SI06: crystal structures of the unbound, affinity-matured Fabs had the same HCDR3 configurations as those in the antigen-bound state; the UCA and I-2, however, were either disordered or constrained, due to crystal packing, in a binding-incompatible state. To determine whether the mutations Y35N and H35N could stabilize a binding-compatible HCDR3 conformation, we determined X-ray crystal structures of unbound Fabs containing Y35N in the UCA background or Y35N and H35N in the I-2 background and compared them to previously determined structures (*Figure 4—figure supplement 1C*). These variants were insufficient to rigidify the HCDR3 as observed by the HCDR3 conformation and high B factors or the lack of density corresponding to the HCDR3 (*Figure 4—figure supplement 1C*). These data show that the I-2 mutations conferred affinity towards MA90-G189E by stabilizing the HCDR1 and HCDR2 with HA and were required for the addition of Y35N and H35N, which remove contacts with HCDR3, to confer affinity against SI06 without complete HCDR3 rigidification, revealing a biophysical mechanism through which inter-chain epistasis can determine broad affinity in a hierarchical manner.

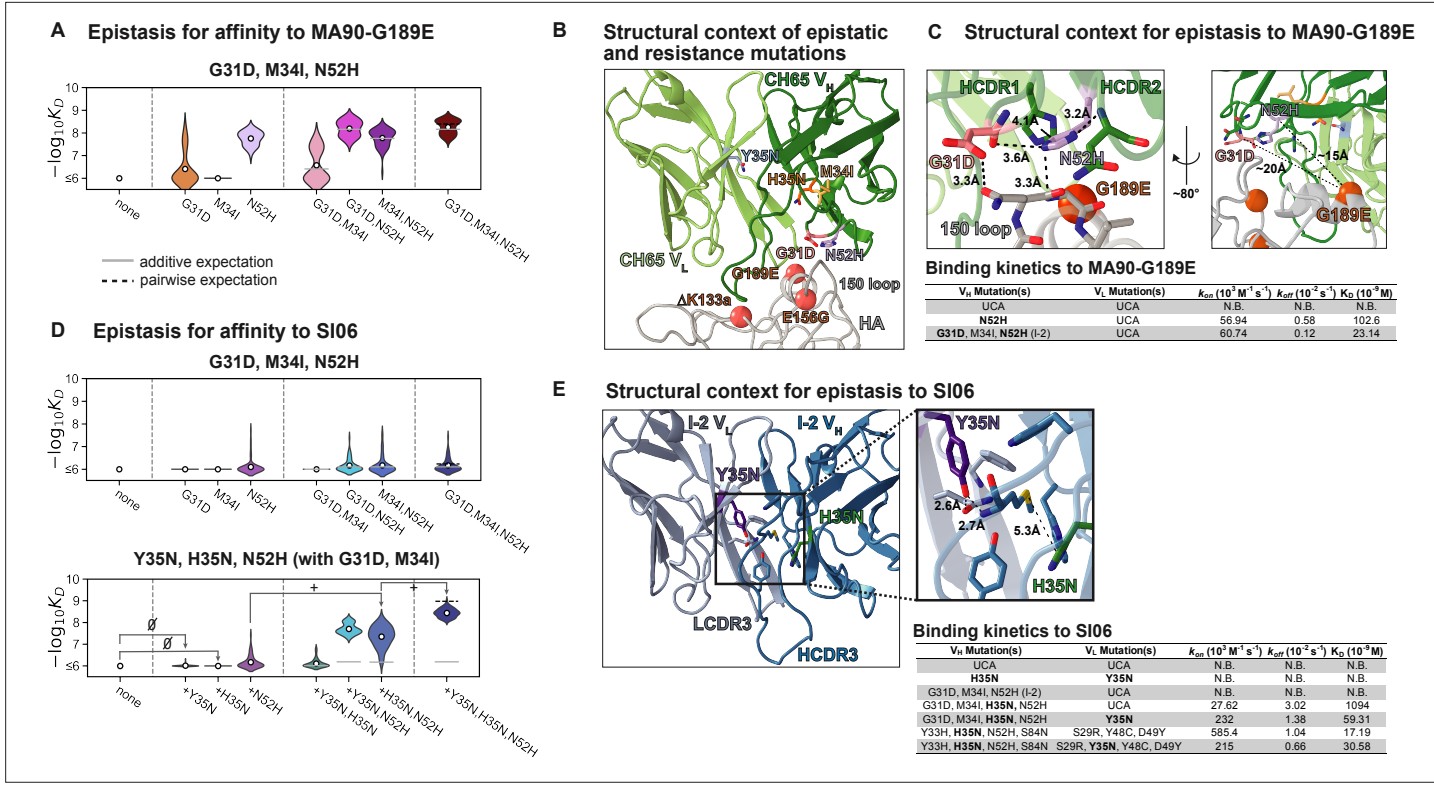

**Figure 4.** Structural basis of epistasis in CH65. (**A**) Three mutations in I-2 (G31D, M34I, N52H) confer affinity to MA90-G189E. Each violin contains 64 genotypes that have the unmutated common ancestor (UCA) residue at positions 35, 48, 49 (V_L) and 33, 35, 85, 87 (V_H), and are variable at the remaining six positions. In (**A**) and (**D**), the white dots indicate the distribution means, and the gray and dotted lines indicate the additive and pairwise expectations, respectively. (**B**) Epistatic mutations that confer affinity to viral escape strains are distant from the sites of escape. Shown is CH65 bound to SI06 (PDB 5UGY; **Whittle et al., 2011**). Colored residues highlight the locations of the mutations shown in (**A**) and (**D**). Spheres highlight the locations of the viral escape mutations (G189E, ΔK133a, and E156G). (**C**) Top, left: mutations N52H and G31D establish a network of interactions between HCDR1, HCDR2, and HA. Top, right: distance between G189E and N52H or G31D precludes interaction. Alpha carbon distances are shown. Bottom: Binding kinetics against MA90-G189E for select variants at 30°C by biolayer interferometry using a bivalent analyte binding model. (**D**) Mutations in I-2 are insufficient for affinity to SI06 (top) but interact epistatically with Y35N and H35N to bind SI06 (bottom). Top: each violin contains 64 genotypes that have the UCA residue at positions 35, 48, 49 (V_L) and 33, 35, 85, 87 (V_H), and are variable at the remaining six positions. Bottom: each violin contains 64 genotypes that have the UCA residue at positions 48, 49 (V_L) and 33, 85, 87 (V_H), the CH65 residue at positions 31 and 34 (V_H), and are variable at the remaining six positions. '∅' and '+' indicate mutations that have neutral or beneficial mean effects on -log$K_D$, respectively. (**E**) Left: epistatic mutations Y35N and H35N are located at the VH-VL interface. Right: somatic mutations remove interactions with the HCDR3. Shown is the unbound I-2 structure (PDB 4HK3 **Schmidt et al., 2013**). Bottom: binding kinetics against SI06 for select variants at 30°C by biolayer interferometry using a bivalent analyte binding model.

The online version of this article includes the following source data and figure supplement(s) for figure 4:

**Source data 1.** Binding kinetics for selected antibody variants determined by biolayer interferometry.

**Source data 2.** X-ray data collection and refinement statistics for unbound Fabs.

**Figure supplement 1.** Structural analysis of HCDR3 conformations observed in crystal structures of unbound and bound Fabs in the CH65 lineage.

**Figure supplement 2.** Representative biolayer interferometry binding traces against MA90 for the indicated antibodies (left) and temperatures (top).

**Figure supplement 3.** Representative biolayer interferometry binding traces against MA90-G189E for the indicated antibodies (left) and temperatures (top).

**Figure supplement 4.** Representative biolayer interferometry binding traces against SI06 for the indicated antibodies (left) and temperatures (top).

**Figure supplement 5.** Summary of all association rates, dissociation rates, and dissociation constants measured by BLI against MA90, MA90-G189E, and SI06 at multiple temperatures.

**Figure supplement 6.** Biolayer interferometry binding traces for the antibody variant containing the I-2 mutations (G31D, M34I, and N52H) in addition to H35N and Y35N for the indicated antigens (left) and temperatures (top).

## Likelihood of mutational pathways to CH65

The extent of epistasis we observe suggests that the evolution of CH65 is contingent on mutations occurring in a particular order. Further, the hierarchical pattern of mutations that confer affinity to the different antigens indicates that the likelihood a mutation fixes depends on the selecting antigen. Because we measured affinities for a combinatorially complete library, we can infer the likelihood of all possible evolutionary trajectories from the UCA to CH65 (with and without the constraint of passing through the I-2 intermediate) in the context of various possible antigen selection scenarios (e.g., maturation to MA90 alone, or to SI06 alone, etc.). To this end, we implement a framework in which the probability of any mutational step is higher if -log$K_D$ increases and lower if -log$K_D$ decreases (see 'Materials and methods'; *Phillips et al., 2021*). We use -log$K_D$ to each antigen to compute the likelihood of all possible mutational trajectories in the context of each of the antigens, as well as in

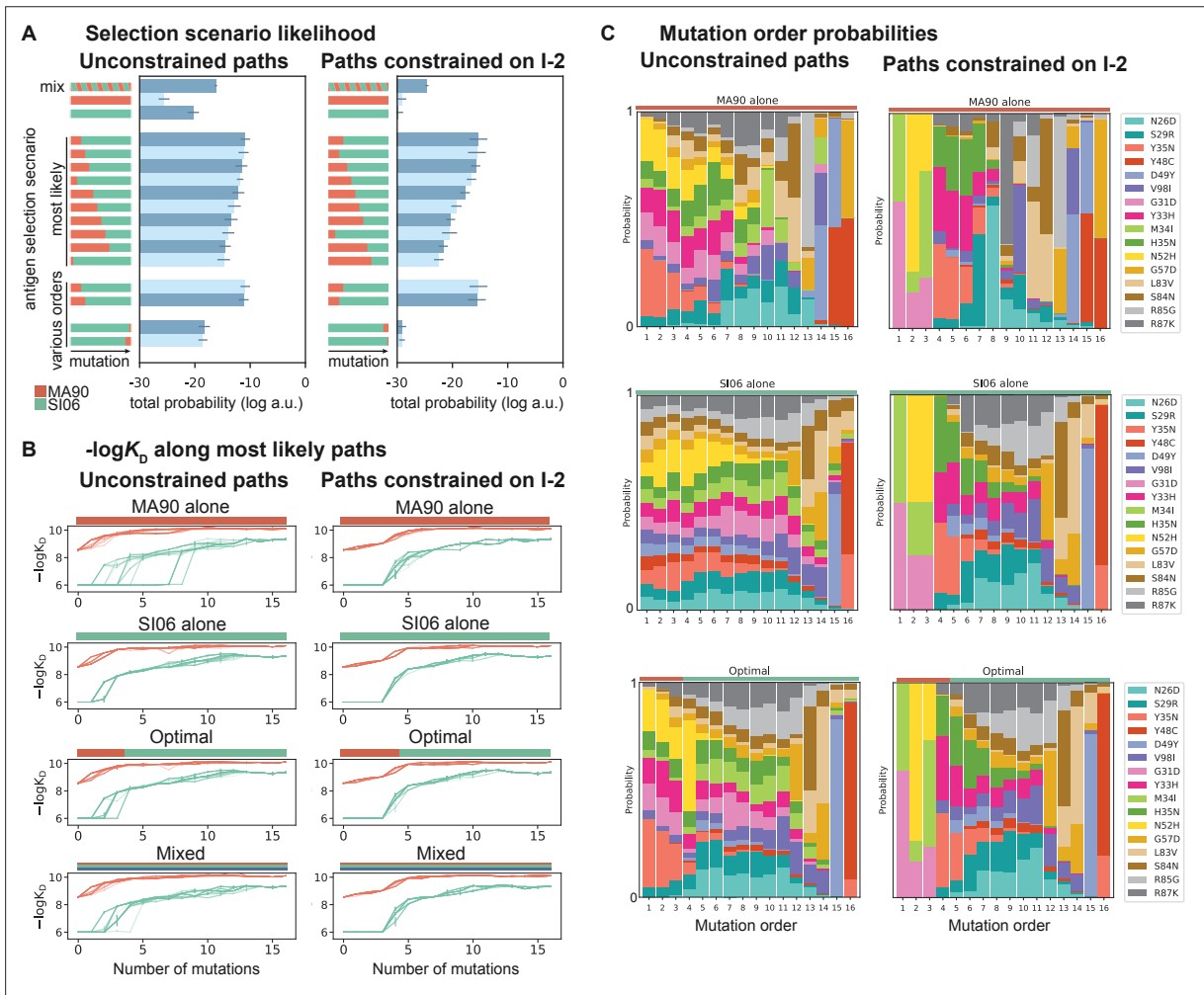

**Figure 5.** Antigen selection scenarios and likely mutational pathways. (**A**) Selection scenario likelihood. Total log probability (in arbitrary units) of all mutational paths from the unmutated common ancestor (UCA) to CH65 (left) or paths from the UCA to CH65 that pass through I-2 (right), assuming specific antigen selection scenarios are shown. 'Most likely' scenarios are those with the highest total probability; 'various orders' show the most likely scenarios for scenarios that begin with MA90 and alternatively, with SI06. Error bars indicate standard error obtained through bootstrap, see 'Materials and methods.' (**B**) -log$K_D$ for 25 most likely paths under designated antigen selection scenarios are shown with (right) and without (left) the constraint of passing through I-2. (**C**) Probability of each mutation occurring at a specific order under select antigen selection scenarios, with (right) and without (left) the constraint of passing through I-2.

The online version of this article includes the following figure supplement(s) for figure 5:

**Figure supplement 1.** Antigen selection scenarios and likely mutational pathways with MA90, SI06, and MA90-G189E.

**Figure supplement 2.** Likelihood of passing through specific 3-mutation intermediates.

**Figure supplement 3.** Graphical model for acquisition of antibody breadth.

the context of all possible sequential selection scenarios, where the selecting antigen can change. We focus on scenarios involving the two antigens that the donor was likely exposed to (*Figure 5*) – MA90 early in their life and SI06 later in their life – but we also perform this analysis with the MA90-G189E data (*Figure 5—figure supplement 1*). Additionally, we consider selection resulting from a mixture of antigens, which we approximate by randomly selecting an antigen for each mutational step (*Wang et al., 2015*) and average this pathway likelihood over 1000 random draws.

This pathway likelihood inference reveals that mutational trajectories leading to CH65 are most favorable in sequential selection scenarios that begin with MA90 and end with SI06, consistent with the donor's likely exposure history (*Schmidt et al., 2013*; *Schmidt et al., 2015a*). This order is preferred regardless of whether paths are constrained to pass through I-2 (*Figure 5A*). Further, the MA90-SI06 sequential scenarios are considerably more likely than either antigen alone, a mixture of antigens, or SI06-MA90 sequential scenarios.

These drastic differences in scenario likelihood result from the effects of specific mutations on various genetic backgrounds. Mutations on the UCA background can improve affinity to MA90 but not to SI06, so MA90 is favored as the selecting antigen initially. After a few mutations, however, MA90 reaches maximal affinity and cannot improve further, at which point mutations begin to improve SI06 affinity. Thus, SI06 is favored later in mutational trajectories (*Figure 5B*). These constraints reflect the structure of the sequence-affinity landscape: selection with MA90 favors mutations that enable the acquisition of SI06 affinity and would be unlikely to occur under selection with SI06 alone. Similarly, when we consider all three antigens, we find that scenarios that begin with MA90 or MA90-G189E and end with SI06 are most likely, again reflecting the hierarchical nature of the sequence-affinity landscape (*Figure 5—figure supplement 1*).

We also leverage our combinatorial data to infer the probability of each mutation occurring at a given step along the evolutionary pathway from UCA to CH65 (*Figure 5C*). Even when we do not constrain pathways to pass through I-2, we find that two of the I-2 mutations (G31D and N52H) and the epistatic mutations that interact with the HCDR3 (e.g., Y33H, H35N, and the previously uncharacterized Y35N) are most likely to occur early in mutational trajectories, especially in scenarios that begin with MA90 selection. Additionally, the highly synergistic HCDR3-stabilizing mutations Y48C and D49Y are most likely to occur late, and consecutively, with D49Y preceding the otherwise-deleterious Y48C. These general trends are robust to constraining paths to pass through the I-2 intermediate. Consistent with our structural analyses, we find that when pathways are constrained to pass through I-2, Y35N is the most probable subsequent mutation, and this likelihood rapidly decreases with additional mutations. However, when we consider all possible pathways in the optimal antigen selection scenario, I-2 is not the most likely 3-mutation intermediate, suggesting that the evolution of CH65 was not contingent on passing through I-2 (*Figure 5—figure supplement 2*). Still, the likelihood of passing through the I-2 intermediate is ~55% higher than that expected by chance – it is a minimally mutated antibody with improved affinity to MA90 and MA90-G189E, and it contains the N52H mutation that is essential for SI06 affinity. Thus, while there are many accessible paths to CH65, the three mutations in I-2 result in rapid improvements in affinity and breadth and favor subsequent selection for epistatic mutations that ultimately provide the breadth of CH65.

## Discussion

Collectively, we find that the breadth of an RBS influenza bnAb, CH65, is determined by high-order epistatic interactions that differ between divergent antigens (*Figure 5—figure supplement 3*). This epistasis is widespread within and between both the heavy and light chains. To our knowledge, this is the first comprehensive study of inter-chain epistasis and illustrates the extent to which mutations can differentially impact affinity depending on the presence of other mutations, even those too far apart to physically interact. This suggests that the maturation of antibodies that engage antigen with both chains may be distinct compared to those that do not. There are more opportunities for epistatic interactions across two chains compared to just one, and the degree of both intra- and inter-chain epistasis is likely contingent on the chain pairing. Given the importance of both light and heavy chain mutations across diverse bnAbs (*Corti et al., 2011*; *Schmidt et al., 2015a*; *Xiao et al., 2019*; *Ekiert and Wilson, 2012*), understanding the nature of this epistasis may be useful for designing therapeutic antibodies and eliciting broadly protective immune responses.

Further, our structural analysis shows that the epistasis that confers broad reactivity in this antibody is mediated through sets of mutations that both interact with HA and those that do not interact with HA. These epistatic mutations add or remove interactions between CDRs, FWRs, and chains that act by different mechanisms of stabilizing the binding conformation (e.g., G31D and N52H) or removing constraints on the HCDR3 (e.g., Y35N and H35N). The Y35N mutation effectively removes interactions between the LFWR2 and the HCDR3 at the $V_H$-$V_L$ interface and mediates affinity improvement to a more antigenically advanced influenza strain by increasing the association rate. An analogous observation was noted for the anti-HIV bnAb CH103 that co-evolved during a natural infection within in a single individual (*Liao et al., 2013*). Structural studies of CH103 identified mutations at the $V_H$-$V_L$ interface (which is the region containing residue 35 in CH65) that were associated with reconfiguration of the HCDR3 to enable broad reactivity against a viral escape variant (*Fera et al., 2014*). Although additional work will be needed to address the generality of this finding, it appears that antibodies can evolve to bind viral escape variants by modulating the $V_H$-$V_L$ interface and the HCDR3 configuration, in response to both chronic (HIV) and punctuated (influenza) exposures. While Y35N is advantageous early in affinity maturation, it becomes detrimental in highly mutated backgrounds that have undergone HCDR3 rigidification. Consequently, Y35N may function to initially increase flexibility, enabling acquisition of affinity to SI06 after acquiring mutations in the heavy chain, and subsequent maturation rigidified the HCDR3. This increased flexibility followed by rigidification is reminiscent of molecular dynamics studies of anti-HIV bnAbs that suggest initial increases in flexibility may provide a means to sample additional conformational space prior to rigidification (*Ovchinnikov et al., 2018*).

In comparing CH65 mutations that improve affinity to diverse H1 antigens, we find that increasingly divergent antigens require additional epistatically interacting mutations, resulting in a hierarchical pattern of mutations that improve affinity to distinct antigens. The I-2 mutations (e.g., G31D, M34I, N52H) may compensate for the G189E mutation by stabilizing interactions with HA opposite this site, potentially allowing the antibody to shift to relieve the clash; a similar observation was made for another RBS-directed antibody (*McCarthy et al., 2019*). These same mutations help to stabilize binding in the antigenically distant SI06 but do not sufficiently compensate for the loss of potential contacts between the HCDR3 and the RBS (e.g., ΔK133a and E156G) within the antigen combining site; further mutations Y35N and H35N that likely influence HCDR3 conformations are needed. These structural observations and the data generated here suggest that mutations confer broad reactivity in the CH65 lineage in a hierarchical manner. Although the hierarchical landscape of CH65 is not as striking as that of CR9114 (*Phillips et al., 2021*), where larger sets of mutations are required to bind substantially more divergent antigens, it is intriguing that the landscape for a considerably narrower bnAb can also have this structure. This suggests that hierarchical sequence-affinity landscapes may be quite common, as they are not unique to CR9114, to anti-stem bnAbs, or to bnAbs that engage distinct HA subtypes.

If hierarchical sequence-affinity landscapes are common amongst bnAbs, they may contribute to the low frequencies of bnAbs in human repertoires. For bnAbs with such landscapes, epistatically interacting mutations are required to bind a given antigen, additional epistatic mutations (that interact favorably with those acquired previously) are required to bind a distinct antigen, and so on. Determining how this might constrain bnAb evolution will require assessing how rare these sets of synergistic mutations are. Importantly, the landscapes measured here and in our previous work focus exclusively on mutations present in the affinity-matured antibodies, which are biased by the selection pressures those bnAbs experienced. Thus, while these landscapes show that diverse bnAbs can mature by acquiring hierarchical sets of epistatic mutations that are favored in sequential exposure regimens, there may be alternative mutational pathways to breadth that are not hierarchical and are favored in other exposure regimes.

Still, the observation that antibodies *can* evolve breadth through hierarchical mutational landscapes lends support for vaccination with sequential doses of distinct antigens. These findings are consistent with a recent study that demonstrates memory B cell recruitment to secondary germinal centers upon vaccination in humans, allowing for additional rounds of antibody maturation to antigenically drifted strains (*Turner et al., 2020*). Here, we find that sequential exposures with antigenically drifted strains may help elicit within-subtype potent bnAbs like CH65, in addition to the cross-subtype bnAbs described in our previous work. Several theoretical and computational models of bnAb affinity maturation also favor sequential immunization strategies as they allow antibodies to acquire the mutations

necessary to bind one antigen before experiencing selection pressure to bind another, ultimately producing bnAbs that have 'focused' on conserved epitopes (*Wang et al., 2015*; *Sachdeva et al., 2020*; *Wang, 2017*; *Molari et al., 2020*; *Sprenger et al., 2020*). Our work indicates that this focusing process may occur by favoring selection on hierarchical sets of epistatically interacting mutations. We note that while the hierarchical epistasis we observe favors the acquisition of breadth to a set of specific antigens, antagonistic epistasis between these mutations and new mutations could prevent the acquisition of breadth to other antigens. Further, both CH65 and CR9114 have higher affinity to the strain most like the inferred original immunogenic stimulus, and weaker affinity to more divergent strains (*Dreyfus et al., 2012*; *Schmidt et al., 2015a*). This is consistent with the concept of immunological imprinting or original antigenic sin, where antibodies boosted upon vaccination or infection typically have high affinity for the eliciting strain (*Guthmiller and Wilson, 2018*). Although we show that the CH65 antibody lineage can evolve breadth that compensates for viral escape mutations, the affinities are lower for more antigenically distant strains, suggesting that there is likely a trade-off between antibody breadth and affinity. Further work will be needed to assess whether RBS-targeting bnAbs like CH65, which target highly variable epitopes compared to stem-targeting bnAbs (*Schmidt et al., 2013*; *Schmidt et al., 2015a*), can mature to bind substantially divergent strains (e.g., post-pandemic H1N1 strains that CH65 does not effectively neutralize), or whether historical contingency prevents them from doing so.

Finally, although epistasis can make evolution more difficult to predict (*de Visser et al., 2018*; *Park et al., 2022*), the general patterns of epistasis emerging from these combinatorial landscapes suggest that there are indeed broadly applicable insights. For example, these hierarchical synergistic interactions reveal how epistasis constrains the evolution of antibody affinity, breadth, and trade-offs between the two. Moving forward, additional combinatorial antibody libraries will advance our understanding of how pervasive these features are – for example, for antibodies that target distinct viruses. Ultimately, though, to understand why we observe these particular bnAbs and not others, we need to explore the unobserved regions of sequence space. We also need to assess the numerous other properties that likely impact selection on antibodies (e.g., stability, folding, polyreactivity). Thus, integrating approaches like this combinatorial approach with methods for assessing local mutational landscapes (e.g., deep mutational scanning) and methods to measure other antibody properties in high throughput will provide a more comprehensive view of the factors that constrain and potentiate antibody evolution.

# Materials and methods

**Key resources table**

| Reagent type (species) or resource | Designation | Source or reference | Identifiers | Additional information |
|---|---|---|---|---|
| Strain, strain background (*Saccharomyces cerevisiae*) | EBY100 | ATCC | Cat# MYA-4941 | |
| Strain, strain background (influenza A virus) | Influenza A/Puerto Rico/8/1934 with A/Massachusetts/1/1990HA and A/Siena/10/1989 NA | Jesse Bloom and this paper | | GenBank: L19027 (HA); GenBank: CY036823 (NA) |
| Cell line (*Homo sapiens*) | HEK293T | ATCC | Cat# CRL-3216 | |
| Cell line (*Canis lupus familiaris*) | MDCK-SIAT1 | MilliporeSigma | Cat# 05071502 | |
| Cell line (*H. sapiens*) | Expi293F | Gibco | Cat# A14527 | |
| Antibody | Anti-cMyc-FITC (mouse monoclonal) | Miltenyi Biotec | Cat# 130-116-485 | FACS (1:50) |
| Recombinant DNA reagent | pCHA (plasmid) | Dane Wittrup *Mata-Fink et al., 2013* | | |
| Recombinant DNA reagent | pCHA_UCA860_scFv (plasmid) | This paper | | Plasmid map in *Supplementary file 1* |

*Continued on next page*

*Continued*

| Reagent type (species) or resource | Designation | Source or reference | Identifiers | Additional information |
|---|---|---|---|---|
| Recombinant DNA reagent | pCHA_CH65_scFv (plasmid) | This paper | | Plasmid map in **Supplementary file 2** |
| Sequence-based reagent | CH65 golden gate primers | IDT | | Sequences listed in **Supplementary file 3** |
| Sequence-based reagent | Illumina sequencing primers | IDT | | Sequences listed in **Supplementary file 4** |
| Peptide, recombinant protein | Streptavidin-RPE | Thermo Fisher | Cat# S866 | FACS (1:100) |
| Peptide, recombinant protein | A/Massachusetts/1/1990 – MA90 | This paper | | Sequence in **Supplementary file 5** |
| Peptide, recombinant protein | A/Massachusetts/1/1990 – MA90-G189E | This paper | | Sequence in **Supplementary file 6** |
| Peptide, recombinant protein | A/Solomon Islands/03/2006 | This paper | | Sequence in **Supplementary file 7** |
| Peptide, recombinant protein | Various Fabs & IgGs | This paper | | See **Figure 4—source data 1** for specific sequences |
| Commercial assay or kit | BirA500 kit | Avidity | | |
| Commercial assay or kit | Zymo Yeast Plasmid Miniprep II | Zymo Research | Cat# D2004 | |
| Software, algorithm | Custom code | This paper | | github.com/amphilli/CH65-comblib |
| Software, algorithm | Interactive data browser | This paper | | https://ch65-ma90-browser.netlify.app/ |

For all methods, 'biological replicates' refer to independent experiments performed on different days, and 'technical replicates' refer to multiple measurements of the same biological sample.

## Antibody library production

### Antibody sequences and mutations of interest

The UCA860 amino acid sequence (**Whittle et al., 2011**) was codon-optimized for expression in yeast. Amino acid substitutions corresponding to those in CH65 were encoded by ≥2 nucleotide mutations, when possible. The V98I mutation, which lies outside the region captured by 2 × 250 bp reads, was encoded by a synonymous mutation at Arg53. The Q1E and S75A mutations in $V_H$ were determined to minimally influence affinity (**Figure 1—figure supplement 1**) and were excluded from all subsequent experiments to reduce the library size.

### Yeast display plasmid and strains

Single-chain variable format (scFv) antibody constructs were cloned via Gibson Assembly (**Gibson et al., 2009**) into the pCHA yeast display vector (**Van Deventer et al., 2015**) with a C-terminal myc epitope tag and Aga-2 fusion (**Supplementary files 1 and 2**). These scFv constructs were displayed on the surface of the EBY100 yeast strain (**Boder and Wittrup, 1997**), as described below for the yeast library production. Unless otherwise noted, yeast were cultured by rotating at 30°C and were pelleted by centrifuging at 14,000 × *g* (1 min) or 3000 × *g* (10 min).

### Combinatorial Golden Gate Assembly

To assemble the combinatorially complete library containing all $2^{16} = 65,536$ variants, the scFv sequence was sectioned into five fragments of roughly equal length such that each fragment contained ≤5 mutations. Primers were designed to create all possible ($≤2^5$) versions of each fragment by adding mutations, a Bsa-I cleavage site, and a 4 bp overhang unique to each fragment (**Supplementary file 3**). Fragments were amplified from the UCA860 sequence via PCR using Q5 Polymerase (NEB, Ipswich, MA, #M0491). The resulting fragments were purified using a 2× ratio of Aline beads (Aline Biosciences, Woburn, MA, #C-1003-5), overnight DpnI digestion at 37°C (NEB #R0176), and a second 2× ratio bead cleanup. The backbone vector was prepared by replacing the scFv sequence in the pCHA

yeast display vector with a *ccdb* counter-selection marker. Equimolar amounts of each fragment were then pooled and assembled into the backbone vector at a 2:1 molar ratio via Golden Gate Assembly (*Engler et al., 2008*; NEB #R3733). The assembly mix was then transformed into electrocompetent DH10B *Escherichia coli* in 5 × 25 uL cell aliquots (NEB #C3020). Each cell aliquot was recovered in 1 mL outgrowth media at 37°C for 1 hr and then transferred into 100 mL of molten LB (1% tryptone, 0.5% yeast extract, 1% NaCl, 100 g/L ampicillin [VWR # V0339], 0.4% SeaPrep agarose [VWR, Radnor, PA #12001-922]) in a 500 mL baffled flask. The bacteria–agar mixture was incubated at 4°C for 3 hr to gel the agar and was then incubated at 37°C for 16 hr. Each flask contained 1–2 million colonies (5–10 million colonies across five flasks; >100 times the library diversity) and was blended by shaking at 200 rpm for 1 hr. The cells were then pelleted by spinning at 3000 × *g* for 10 min, and plasmid DNA was extracted using the ZymoPURE II Plasmid Midiprep Kit (Zymo Research, Irvine, CA, #D4201).

## Yeast library production

One day prior to transformation, EBY100 cells were thawed by inoculating 5 mL YPD (1% Bacto yeast extract [VWR #90000-726], 2% Bacto peptone [VWR #90000-368], 2% dextrose [VWR #90000-904]) with 150 µL glycerol stock and rocking at 30°C for 12–24 hr. The scFv plasmid library was then transformed into EBY100 cells by the lithium acetate method (*Gietz and Schiestl, 2007*) and transformants were recovered in 100 mL molten SDCAA (1.71 g/L YNB without amino acids and ammonium sulfate [Sigma-Aldrich, St. Louis, MO, #Y1251], 5 g/L ammonium sulfate [Sigma-Aldrich #A4418], 2% dextrose [VWR #90000-904], 5 g/L Bacto casamino acids [VWR #223050], 100 g/L ampicillin [VWR # V0339], 0.4% SeaPrep agarose [VWR #12001-922]) in 500 mL baffled flasks. The yeast–agar mixture was incubated at 4°C for 3 hr to allow the agar to set and was then incubated at 30°C for 48 hr to allow for yeast colony growth. Each flask contained ~700,000 colonies, totaling about 7 million colonies across ten flasks (>100 times the library diversity). After disrupting the agar by shaking at 200 rpm for 1 hr, the yeast library was inoculated into liquid SDCAA (1.71 g/L YNB without amino acids and ammonium sulfate [Sigma-Aldrich #Y1251], 5 g/L ammonium sulfate [Sigma-Aldrich, #A4418], 2% dextrose [VWR #90000-904], 5 g/L Bacto casamino acids [VWR #223050], 100 g/L ampicillin [VWR # V0339], 5.4 g $Na_2HPO_4$ [Sigma-Aldrich, #S7907], 8.56 g $NaH_2PO_4.H_2O$ [Sigma-Aldrich, #S9638]) (*Chao et al., 2006*) and grown for five generations to saturation before freezing at –80°C in 1 mL aliquots containing 5% glycerol.

## Viral escape

### Cell lines and media

HEK293T cells (ATCC #CRL-3216; authenticated by STR profiling and verified mycoplasma-negative by manufacturer) were passaged in DMEM (Gibco, #11965126) supplemented with 10% fetal bovine serum (Peak Serum) and Penicillin-Streptomycin (Gibco, #15140163) subsequently referred to as 'D10.' MDCK-SIAT1 cells (Sigma, #05071502; authenticated by STR profiling and verified mycoplasma-negative by manufacturer) were passaged in D10 additionally supplemented with 1 mg/ml Geneticin (Gibco, #10131035). Prior to infection, Geneticin was not included in the MDCK-SIAT1 medium. Media used to propagate influenza, referred to as 'flu media,' contain Opti-MEM (Gibco, #31985088) supplemented with 0.3% BSA (Roche, #03117332001), 0.01% FBS, and Penicillin-Streptomycin. Prior to propagation, 1 µg/mL of TPCK-trypsin (Sigma, #T1426) was freshly added to flu media.

### Generation of recombinant MA90 virus

We used a standard eight plasmid reverse genetics system (*Hoffmann et al., 2000*) to generate a recombinant 6:2 virus bearing the PB2, PB1, PA, NP, M, and NS genomic segments from PR8 (A/Puerto Rico/8/1934; a kind gift from Jesse Bloom), MA90 HA (GenBank: L19027), and A/Siena/10/1989 NA (GenBank: CY036825). Because the sequencing of the MA90 HA was not complete, the C-terminus was extended with that of A/Siena/10/1989 (GenBank: CY036823). In a six-well plate treated with poly-L-lysine (Sigma, #P4707), 6 × 10$^5$ HEK293T cells and 1 × 10$^5$ MDCK-SIAT1 cells were added to wells six-well plates in D10. The next day, media was aspirated from the cells and fresh, pre-warmed D10 was added on top. For each transfection, 8 µL of Trans-IT LT1 (Mirus, #2300) was added to Opti-MEM (Gibco, #31985070) containing 0.5 µg of each plasmid and incubated at room temperature for 20 min. The mixture was then added dropwise to the cells. After ~5 hr, the media was aspirated from the cells and flu media freshly supplemented with 1 µg/mL TPCK-treated trypsin was added.

After 2 days, dead cells were removed from the virus-containing media by centrifugation at 800 × *g* for 5 min. The supernatant was then supplemented with 1 μg/mL TPCK-treated trypsin and added to a confluent monolayer of MDCK-SIAT1 cells seeded 1 day before in a six-well plate and washed once with PBS (seeded at 7 × 10^5 cells per well). After ~4–5 hr, the supernatant was removed, and fresh flu media supplemented with 1 μg/mL TPCK-treated trypsin was added. One day later, successful rescue was judged by observing cytopathic effect. Multiple rescue transfections were pooled and added to 10 cm dishes containing a confluent monolayer of MDCK-SIAT1 cells seeded 1 day prior (at 3 × 10^6 cells per dish) as detailed above. Two days later, successful propagation was judged by cytopathic effect, the supernatant was clarified by centrifugation, and aliquots were frozen at –80°C.

## Escape variant generation

Prior to infection, MA90 virus was incubated with a low concentration of antibody (started at 0.01 μg/mL of the UCA), a higher concentration of antibody (one half-log greater than the lower concentration), or no antibody (as a control for cell line adaptation mutations) in 500 μL of flu media supplemented with 1 μg/mL TPCK-treated trypsin for 1 hr at 37°C and 5% $CO_2$. MDCK-SIAT1 cells seeded the day before were washed with PBS and then virus–antibody mixtures were added to the monolayers and incubated for 1 hr at 37°C and 5% $CO_2$, rocking the plate every ~15 min to ensure that the cells did not dry out. Afterward, the viral inoculum was removed, and the cells were washed with PBS before adding fresh flu media supplemented with 1 μg/mL TPCK-treated trypsin. After 2 days, viral growth was judged by cytopathic effect. The well that grew with a higher concentration of antibody was selected for the next passage where the 'low' antibody concentration was the same as the previous passage and the 'high' concentration was a half-log higher. This process was repeated until viral growth was readily detectable at 100 μg/mL of the UCA. If necessary, a hemagglutination assay using turkey red blood cells (Lampire, #7249409) was run to determine whether virus was present. Briefly, twofold dilutions of the virus in PBS were mixed with 0.5% turkey red blood cells and incubated at room temperature for at least 30–45 min before visualization of red blood cell pellets to determine whether virus had grown significantly. Once the virus still grew in 100 μg/mL of the antibody, the virus was passaged one additional time and 100 μg/mL of antibody was additionally added to the media added after infection. The RNA from the escaped virus was isolated using a QIAamp viral RNA mini kit (QIAGEN, #52904), and the full-length HA was amplified using gene-specific primers and the OneStep RT-PCR kit (QIAGEN, #210212). The resulting PCR product was sequenced by Sanger sequencing (Genewiz). The mutation G189E was identified from the sequencing results and produced as a recombinant protein for subsequent experiments (see below).

## Antigen and IgG production

### Choice of HA antigens

Antibodies CH65, CH66, and CH67 were isolated from plasmablasts from donor TIV01 (*Moody et al., 2011*) after receiving the trivalent influenza vaccine in the 2007–2008 influenza season, which contained the A/Solomon Islands/3/2006 (SI06) H1N1 strain. The donor TIV01 was born in ~1990 and subsequent work identified that the inferred UCA of this lineage bound to the strain A/Massachusetts/1/1990 (MA90) circulating near the donor's birth date and is suspected to be highly similar to the original immunogenic stimulus of this lineage (*Schmidt et al., 2015a*). However, the UCA did not bind SI06, which escaped the UCA and I-2 of this lineage (*Schmidt et al., 2015a*). To assess whether affinity maturation in this lineage is capable of accommodating for an escape mutation that abrogates binding to less mature variants, we drove viral escape from MA90 in vitro (see above) using the UCA and identified that matured variants of this lineage (e.g., CH65 and CH67) bound the escape variant (MA90-G189E) with high affinity. To understand how this antibody lineage evolved to compensate for viral escape mutations, we included MA90-G189E and SI06 in addition to MA90.

### Recombinant protein cloning, expression, and purification

Variable heavy and light chains were synthesized as eBlocks (IDT). Full-length, codon-optimized HAs (A/Massachusetts/1/1990 – MA90 [*Supplementary file 5*], MA90-G189E [*Supplementary file 6*], and A/Solomon Islands/03/2006 – SI06 [*Supplementary file 7*]) and full-length human IgG1 heavy and light chains were cloned into a pVRC expression vector containing a C-terminal HRV 3C cleavage site, His tag, FoldOn trimerization domain, and AviTag for HAs and a HRV 3C cleavage site followed by a

C-terminal His tag for antibody heavy chains. Recombinant proteins were produced in Expi293F cells (Gibco, #A14527; authenticated by STR profiling and verified mycoplasma-negative by the manufacturer) following the manufacturer's directions. The trimeric HAs were purified from the supernatant using TALON metal affinity resin (Takara, #635653), washing with PBS, and eluting with PBS containing 200 mM imidazole (pH 7.4). After concentration, proteins were further purified over an S200 column on an AKTA pure (Cytiva). For yeast surface display assays, the HAs were further biotinylated and flash-frozen in liquid nitrogen (see below). For kinetics measurements, the HAs were used within 2 weeks of production and never frozen.

## HA biotinylation

Biotinylation of the HAs was performed using the BirA500 kit (Avidity) following the manufacturer's instructions. To compensate for the reduced activity in PBS, twice the amount of BirA was added and the reaction was additionally supplied with twice the amount of biotin using the supplied BIO-200. The biotinylation reaction was allowed to proceed for 1.5 hr at 30°C before 0.2 µm filtering and purification over an S200 column (Cytiva). The trimeric HAs were then concentrated and flash-frozen in liquid nitrogen for single-use aliquots. Biotinylated HAs were quality controlled by a gel shift assay. Approximately 2 µg of biotinylated HA was heated in non-reducing Laemmli buffer (Bio-Rad, #1610737) at 95°C for 5 min. Once cooled to room temperature, excess streptavidin was added and allowed to incubate for at least 5 min. As a control, samples were run with PBS added rather than streptavidin. The mixture was then run on a Mini-PROTEAN TGX Stain-Free gel (Bio-Rad, #4568096) and imaged. All biotinylated HAs shifted in the presence of streptavidin, indicating successful biotinylation.

## Tite-Seq assays

Tite-Seq assays were performed in biological duplicate (on different days) for each antigen, as previously described (*Phillips et al., 2021*; *Adams et al., 2016*) with some modifications described below.

### Induction of antibody expression

On day 1, the yeast CH65 library and isogenic strains containing the pCHA-UCA860 or pCHA-CH65 plasmids were thawed by inoculating 5 mL SDCAA with 150 µL glycerol stock and rotating at 30°C for 24 hr. On day 2, yeast cultures were back-diluted to OD600 = 0.2 in 5 mL SDCAA and rotated at 30°C until they reached an OD600 = 0.4–0.6 (about 4 hr). Subsequently, 1.5 mL of these log-phase cultures were pelleted, resuspended in 4 mL SGDCAA (1.71 g/L YNB without amino acids and ammonium sulfate [Sigma-Aldrich #Y1251], 5 g/L ammonium sulfate [Sigma-Aldrich, #A4418], 1.8% galactose [Sigma-Aldrich #G0625], 0.2% dextrose [VWR #90000-904], 5 g/L Bacto casamino acids [VWR #223050], 100 g/L ampicillin [VWR # V0339], 5.4 g $Na_2HPO_4$ [Sigma-Aldrich, #S7907], 8.56 g $NaH_2PO_4$.$H_2O$ [Sigma-Aldrich, #S9638]) (*Chao et al., 2006*), and rotated at room temperature for 20–22 hr.

### Primary antigen labeling

On day 3, following induction of scFv expression, cultures were pelleted, washed twice with cold 0.1% PBSA (VWR #45001-130, GoldBio, St. Louis, MO, #A-420-50), and resuspended to an OD600 of 1. For each concentration of antigen (0.75-log increments spanning 1 µM to 1 pM), 700 µL of the CH65 yeast library (OD600 = 1) were incubated with biotinylated HA by rocking at 4°C for 24 hr. Notably, the volume of each antigen concentration was adjusted such that the number of antigen molecules exceeded that of antibody molecules by at least tenfold (assuming 50,000 scFv/cell) (*Boder and Wittrup, 1997*).

### Secondary fluorophore labeling

On day 4, yeast-HA complexes were pelleted at 4°C and washed twice with 5% PBSA + 2 mM EDTA. Complexes were then incubated with Streptavidin-RPE (1:100, Thermo Fisher Scientific, Waltham, MA, #S866) and anti-cMyc-FITC (1:50, Miltenyi Biotec, Somerville, MA, #130-116-485) at 4°C for 45 min in the dark. Following incubation, complexes were washed twice with 5% PBSA + 2 mM EDTA and stored on ice in the dark until sorting.

## Sorting

Yeast-HA complexes were sorted on a BD FACS Aria Illu equipped with an 85 micron fixed nozzle and 405 nm, 440 nm, 488 nm, 561 nm, and 635 nm lasers. Single-color controls were used to compensate for minimal overlap between the FITC and PE channels. For all sorts, single cells were gated by FSC vs SSC, and the resulting population was sorted either by expression (FITC) or HA binding (PE). For the expression sort, ~1.6 million (~20× library diversity) single cells were sorted into four gates of equal width spanning the FITC-A axis. For the HA binding sort, ~1.6 million scFv-expressing cells were sorted into four gates spanning the PE-A axis, with one gate capturing all PE-negative cells, and the remaining three each capturing 33% of the PE-positive cells (*Figure 1—figure supplement 2*). All cells were sorted into 5 mL polypropylene tubes containing 1 mL of 2× SDCAA supplemented with 1% BSA and were stored on ice until recovery.

## Recovery and plasmid extraction

Following sorting, yeast were pelleted by spinning at $3000 \times g$ for 10 min at 4°C. Supernatant was carefully removed by pipette, and the resulting pellet was resuspended in 4 mL SDCAA and transferred to a glass culture tube. A small amount of this resuspension (targeting 200–500 cells, based on sorting counts) was plated on SDCAA-agar and YPD-agar to quantify recovery efficiency and plasmid loss. Cultures were then rocked at 30°C until reaching OD600 = 0.8–2.

After reaching the target OD600, 1.5 mL yeast culture was pelleted and frozen at –80°C for at least an hour. Plasmid was then extracted using the Zymo Yeast Plasmid Miniprep II kit (Zymo Research #D2004) following the manufacturer's instructions, except for the following changes: 5 µL zymolyase was used per sample, zymolyase incubations were 2–3 hr, precipitate following neutralization was removed by centrifugation at $21,000 \times g$ for 10 min, columns were washed using 650 µL wash buffer and dried by spinning at $16,000 \times g$ for 3 min, and plasmid was eluted in 15 µL elution buffer.

## Sequencing library preparation

ScFv amplicon sequencing libraries were then prepared by a two-step PCR as previously described (*Nguyen Ba et al., 2019*). The first PCR appended unique molecular identifiers (UMI), sample-specific inline indices, and a partial Illumina adapter to the scFv sequence, and was performed for five cycles to minimize PCR amplification bias. The second PCR appended the remainder of the Illumina adapter and sample-specific Illumina i5 and i7 indices, and was performed for 35 cycles to produce a sufficient amount of each amplicon library (primer sequences in *Supplementary file 4*). The first PCR used 5 µL plasmid DNA as template for a 20 µL reaction using Q5 polymerase according to the manufacturer's instructions with the following cycling program: (1) 60 s at 98°C, (2) 10 s at 98°C, (3) 30 s at 67°C, (4) 60 s at 72°C, (5) GOTO 2, 4×, and (6) 60 s at 72°C. The product from PCR 1 was then brought up to 40 µL with MBG water, purified using Aline beads at a ratio of 1.2×, and eluted in 35 µL elution buffer. 33 µL of this elution was used as template for the second PCR, which was a 50 µL reaction using Kapa polymerase (Kapa Biosystems, Wilmington, MA, #K2502) as per the manufacturer's instructions and the following cycling program: (1) 30 s at 98°C, (2) 20 s at 98°C, (3) 30 s at 62°C, (4) 30 s at 72°C, (5) GOTO 2, 34×, and (6) 300 s at 72°C. The resulting amplicons were purified using Aline beads at a ratio of 0.85×, and DNA concentration was determined using a fluorescent DNA-binding dye (Biotum, Fremont, CA, #31068) as per the manufacturer's instructions. Amplicons were then pooled amongst the four bins for each concentration, based on the number of cells sorted into each gate, and then equimolar amounts of the resulting pools were combined to make the final pooled library. Prior to sequencing, the pool concentration was determined by Qubit and the size verified by Tapestation HS DNA 5000 and 1000. The pool was then sequenced on a NovaSeq SP (2x250 paired-end reads) with 10% PhiX spike-in; 2–4 curves were loaded onto a single flow cell to sequence each variant at at least 100× coverage.

## Sequencing data processing

Demultiplexed sequencing reads were parsed using a Snakemake pipeline as previously described (*Moulana et al., 2022*) (see github.com/amphilli/CH65-comblib for parameters). Briefly, UMI, inline indices, and genotypes were extracted from each read using (*Friedl, 2009*). Reads with incorrectly paired inline indices or unexpected mutations at the CH65 mutation sites were discarded. In all other

regions of the read, all reads exceeding a 10% error rate were discarded. Following this filtering, reads were deduplicated by UMI to generate unique counts files for each sample.

## Tite-Seq $K_D$ inference

### Mean-bin approach

To fit the dissociation constant ($K_D$) for each variant in the library, we followed the same method as previously described (*Phillips et al., 2021*). Briefly, we use the sequencing counts and flow cytometry data to infer the mean log-fluorescence of each genotype $s$ at each concentration $c$:

$$\bar{F}_{s,c} = \sum_b F_{b,c}\, p_{b,s|c}$$

where $F_{b,c}$ is the mean log-fluorescence of bin $b$ at concentration $c$, and $p_{b,s|c}$ is the proportion of cells with genotype $s$ sorted into bin $b$ at concentration $c$, and is given by:

$$p_{b,s|c} = \frac{\frac{R_{b,s,c}}{\sum_s R_{b,s,c}} C_{b,c}}{\sum_b \left( \frac{R_{b,s,c}}{\sum_s R_{b,s,c}} C_{b,c} \right)}$$

where $R_{b,s,c}$ is the number of reads with genotype $s$ found in bin $b$ at concentration $c$, and $c_{b,c}$ is the number of cells sorted into bin $b$ at concentration $c$.

Uncertainty is then propagated in these mean bin estimate as:

$$\delta\bar{F}_{s,c} = \sqrt{\sum_b \left( \delta F_{b,c}^2\, p_{b,s|c}^2 + F_{b,c}^2\, \delta p_{b,s|c}^2 \right)}$$

where $\delta F_{b,c}$ is the standard deviation of log-fluorescence for cells sorted into bin $b$ at concentration $c$. This is approximated by $\sigma F_{b,c}$ and the error in $p_{b,s|c}$ results from sampling error, which is approximated as a Poisson process at sufficient sequencing coverage, yielding:

$$\delta p_{b,s|c} = \frac{p_{b,s|c}}{\sqrt{R_{b,s,c}}}$$

The dissociation constant, $K_{D,s}$, was inferred for each genotype by fitting the logarithm of the Hill function to the mean log-fluorescence:

$$\bar{F}_{s,c} = \log_{10}\left( \frac{c}{c+K_{D,s}} A_s + B_s \right)$$

where $A_s$ is the increase in fluorescence at antigen saturation and $B_s$ is the background fluorescence in the absence of antigen. The fit was performed using the Python package *scipy.optimize curve_fit* function using the following boundary conditions: $A_s$ ($10^2 - 10^6$), $B_s$ ($1 - 10^5$), $K_{D,s}$ ($10^{-14} - 10^{-5}$).

### Data quality and filtering

Following the $K_{D,s}$ inference, non-binding sequences with $K_{D,s}<6$ or $A_s - B_s <1$ were pinned to the titration boundary with -log$K_{D,s}$ = 6. Subsequently, $K_{D,s}$ values resulting from poor fits ($r^2 < 0.8$, $\sigma > 1$) were removed from the dataset, $K_{D,s}$ were averaged across biological replicates, and $K_{D,s}$ with large SEM (>0.5 log units) were excluded from subsequent analyses. This filtering retained 65,530, 63,840, and 64,619 genotypes for the MA90, G189E, and SI06 Tite-Seq experiments, respectively (*Figure 1—source data 1*).

### Expression data

Sequencing reads corresponding to the expression sort were handled identically to those from the HA binding sort, and the mean log-fluorescence was inferred as detailed above. Day-to-day variation in fluorophore labeling and detection were accounted for by normalizing mean log-fluorescence values by the average mean log-fluorescence of the corresponding biological replicate (*Figure 1—source data 1*).

## Force-directed layouts

To reduce the dimensionality of the sequence-affinity landscape, we implemented a force-directed layout, as previously described (**Phillips et al., 2021**). In these graphs, each variant sequence is represented by a node, and variants related by a single mutation are connected by an edge. Edge weights between nodes *s* and *t* are weighted by the change in binding affinity resulting from the corresponding mutation:

$$w_{s,t} = \frac{1}{0.01 + \left| \log_{10} K_{D,s} - \log_{10} K_{D,t} \right|}$$

To construct the force-directed layout, we use $K_{D,s}$ to MA90 to compute the weights. If a mutation from sequence s to t does not impact $K_{D,s}$, those nodes will be close together, and vice versa. The layout coordinates for each variant were obtained using the Python package iGraph function layout_drl, and each node is associated with the corresponding $K_{D,s}$ to SI06 and G189E, as well as the mean expression. An interactive form of this graph is available as an online data browser at https://ch65-ma90-browser.netlify.app/.

## Epistasis analysis

### Linear interaction models

We infer epistatic coefficients as previously described (**Phillips et al., 2021**). Briefly, we implement linear models to infer specific mutational effects and interactions that sum to the observed log-transformed binding affinities, -log($K_{D,s}$), which are proportional to free energy changes and hence expected to be additive (**Wells, 1990**; **Olson et al., 2014**). This additive model is given by:

$$y_s = \beta_0 + \sum_{i=1}^{L} \beta_i x_{i,s} + \varepsilon$$

where *L* is the number of mutations in CH65 (i.e., 16), $\beta_0$ is an intercept, $\beta_i$ is the effect of mutation at site, *i*, $x_{i,s}$ is the genotype of variant *s* at site *i*, and $\varepsilon$ represents independently and identically distributed errors. Our general epistatic model is thus given by

$$y_s = \beta_0 + \sum_{i} \beta_i x_{i,s} + \sum_{i<j}^{L} \beta_{ij} x_{i,s} x_{j,s} + \sum_{i<j<k}^{L} \beta_{ijk} x_{i,s} x_{j,s} x_{k,s} + \ldots + \varepsilon$$

where $\beta_{ij}$ are second-order interaction coefficients between sites *i* and *j*, $\beta_{ijk}$ are third-order interaction coefficients between sites *i*, *j*, and *k*, and so on, up to a specified maximum order of interaction.

We infer these coefficients in both the biochemical and statistical bases (**Phillips et al., 2021**; **Poelwijk et al., 2019**), which are equivalent frameworks related by a linear transformation. For ease of interpretation, we report coefficients inferred using the biochemical model in the main text and figures as these coefficients can be interpreted as mutational effects and interactions relative to the UCA860 sequence. We report coefficients inferred using the statistical model in the figure supplements, and these mutational effects and coefficients can be interpreted as relative to the average of the dataset *Figure 3—figure supplement 4*.

For both biochemical and statistical models, we take a conservative approach to estimating higher-order epistasis. To this end, we truncate the model above some maximal order n and fit the resulting model using a Ridge L2 regularization, beginning with n = 1 and proceeding with higher *n* until the optimal performing model has been identified. We evaluate performance using a cross-validation approach. For each of eight random folds, we use 90% of the data to train the model and evaluate the model using the prediction performance ($R^2$) on the remaining 10%. We then average performance across the eight folds, select the order that maximizes the prediction performance, and retrain the entire dataset on a model truncated at this optimal order, this time by ordinary least-squares regression. This inference yields models with *p* coefficients, and we find that for each antigen $P < N$ by an order of magnitude, where *N* is the number of data points, giving us confidence that we are not overfitting the data.

Practically, we perform this inference using the Python package stats models using ordinary least-squares regression. This yields the coefficient values and associated standard errors and 95% confidence intervals (*Figure 3—source data 1*); coefficients with 95% confidence intervals that do not

cross zero are considered significant and are plotted in *Figure 3* and *Figure 3—figure supplements 2–3*. For the SI06 data, we exclude N52H from the epistasis inference and perform the analysis on the remaining 15 mutations as >90% of sequences with any detectable binding affinity include mutation N52H and thus we do not have power to infer the effect of this mutation. In the statistical epistasis inference, the coefficients at different orders are statistically independent and so we partition the variance explained by the model for each interaction order (*Figure 3—figure supplement 6*).

## Structural analysis of epistasis

To examine the structural context of the first-order and pairwise coefficients from the biochemical epistasis model, we performed two analyses using the co-crystal structure of CH65 with full-length Influenza A/Solomon Islands/3/2006 HA (PDB 5UGY; *Whittle et al., 2011*). First, we used ChimeraX (*Pettersen et al., 2021*) to compute the buried surface area between each mutation in CH65 and HA using the *measure buriedarea* function and the default *probeRadius* of 1.4 Angstroms. This area is plotted as the 'HA contact surface area' in *Figure 3*. We perform the same computation between each mutation in CH65 and the HCDR3, and plot this as the 'CDR3 contact surface area' in *Figure 3*. Second, we used PyMol (The PyMOL Molecular Graphics System, version 2.0 Schrödinger, LLC) to compute the distance between alpha-carbons, and plot this as a function of the pairwise interaction terms in *Figure 3—figure supplement 1*.

## Pathway analysis

### Selection models

To assess the likelihood of mutational pathways from UCA860 to CH65, we assume a moderate selection model in the weak-mutation strong-selection regime as previously described (*Phillips et al., 2021*). Briefly, in this model, mutations fix independently of each other, and mutations are favored if they improve affinity, though both neutral and deleterious mutations are allowed. We use this model to compute the fixation probability of a mutation from sequence $s$ to $t$ (*Kuraoka et al., 2016Kimura, 1962*). This fixation probability is then used to compute the transition probability of the corresponding mutational step:

$$p_{step}\left(\sigma, N\right) = \frac{1 - e^{-\sigma}}{1 - e^{-N\sigma}}$$

We define the selection coefficient $\sigma$ to be proportional to the difference in -log$K_D$ for a particular antigen between sequences $s$ and $t$:

$$\sigma = \gamma\Delta_{s,t}^{ag} = \gamma\left(-\log_{10} K_{D,t}^{ag} - \left(-\log_{10} K_{D,s}^{ag}\right)\right)$$

where $N$ is the effective population size and $\gamma$ corresponds to the strength of selection. For the moderate selection model applied here, we use $N = 1000$ and $\gamma = 1$. Additionally, to compute the total number of mutational paths that improve in affinity at each step, we use $N \to$ infinity and $\gamma \to$ infinity such that $p_{step} = 1$ if the mutation improves affinity and $p_{step} = 0$ otherwise. These fixation probabilities are then used to compute the transition probability for all sequences $s,t$ over all antigens $ag$:

$$P_{s,t}^{ag} = \begin{cases} p_{step}\left(\Delta_{s,t}^{ag}, \gamma, N\right), & \text{if t has one more mutation than s} \\ 0, & \text{otherwise} \end{cases}$$

### Antigen selection scenario likelihood and mutation probabilities

The transition probabilities described above were used to compute the total probability for a set of possible antigen selection scenarios, and for select antigen selection scenarios, the probability of each mutation occurring at a specific order (*Figure 5*). This was performed as previously described (*Phillips et al., 2021*), where the probabilities $P_{s,t}^{ag}$ are stored as sparse transition matrices $P^{ag}$ of dimension $2^N \times 2^N$ for each antigen, where entries are nonzero when sequence $t$ has one more mutation than sequence $s$. To evaluate the total probability for a given antigen selection scenario, we compute the matrix product for all mutational steps $i$ under a specific sequence of antigen selection contexts $ag_1$, …, $ag_L$:

$$P_{tot} = \sum_{paths} \left( \prod_{steps} P_{step} \right) = \left[ \prod_{i=1}^{L} P^{ag_i} \right]_{s_g, s_s}$$

where $[.]_{s,s'}$ represents the matrix element in the row corresponding to genotype $s$ and the column corresponding to genotype $s'$. Notably, transition probabilities are not normalized at each step. Thus, many pathways will not reach the somatic CH65 sequence and the likelihood assesses the probability of reaching the CH65 somatic sequence.

Here, we consider three classes of antigen selection scenarios. The simplest is a single-antigen selection scenario, in which all steps $i$ use the same antigen. Second, we consider selection scenarios where steps can use different antigens in a non-repetitive manner. Finally, we consider a scenario that approximates exposure to a mixture of antigens (**Wang et al., 2015**; **Wang, 2017**; **Kuraoka et al., 2016**), in which an antigen is drawn at random for each mutational step $i$. We then calculate $P_{tot}$ for 1000 randomly drawn scenarios, report the average log probability, and illustrate mutational paths and orders for a scenario near the median probability from the 1000 draws. For all antigen selection scenarios, the error of $P_{tot}$ is estimated by resampling the binding affinity from a normal distribution corresponding to its value and standard deviation. We perform this bootstrapping over 10 iterations and report $P_{tot}$ as the average.

To identify the most likely paths under a given selection scenario (as plotted in **Figure 5B**), we construct a directed graph, where each sequence $s$ is a node, and edges connect nodes $s$ and $t$ that are separated by one mutation. The edge weights are calculated from the transition probability, $w_{s \rightarrow t} = -\log \left( P_{s,t}^{ag} + \epsilon \right)$. In this framework, we can use the *shortest_simple_paths* function in Python package *networkx* (**Hagberg et al., 2022**) to compute the most likely paths.

To calculate the probability that a mutation at site $m$ happened at a specific step $j$, we normalize the transition matrix (i.e., all paths must reach the somatic CH65 sequence) for a given antigen selection context:

$$\widetilde{P}_{s,t}^{ag} = P_{s,t}^{ag} \times \left( \sum_{t} P_{s,t}^{ag} \right)^{-1}$$

For $P_{s,t}^{ag} \neq 0$ and 0 otherwise. The total relative probability for that site mutating at a specific step under an antigen exposure scenario is given by

$$P_{j,\alpha} = \left[ \left( \prod_{i=1}^{j-1} \widetilde{P}^{ag_i} \right) \cdot \widetilde{P}_{\alpha}^{ag_j} \cdot \left( \prod_{i=j+1}^{L} \widetilde{P}^{ag_i} \right) \right]_{s_g, \, s_s}$$

Finally, to determine the total probability of each variant (**Figure 5—figure supplement 2**), which is given by the sum of the probabilities of all paths passing through that variant in a specific antigen selection scenario:

$$P_s = \left( \left[ \prod_{i=1}^{j} \widetilde{P}^{ag_i} \right]_{s_g, \, s} \right) \cdot \left( \left[ \prod_{i=j+1}^{L} \widetilde{P}^{ag_i} \right]_{s, s_s} \right)$$

where $j$ is the number of somatic mutations in variant $s$, the first term is the probability of reaching sequence $s$ at mutational step $j$, and the second term is the probability of reaching the CH65 sequence after passing through sequence $s$. We perform an additional normalization, $P_s' = P_s \times n_j$, so that variants with different numbers of mutations can be compared. $P_s'$ is thus the probability of a specific variant in the selective model compared to a neutral model (e.g., sequences with $\log(P_s') > 0$ are favored).

## Isogenic $K_D$ measurements

To validate $K_D$ measurements made using Tite-Seq, we generated isogenic yeast strains encoding select variants in the CH65 scFv library and measured their affinity to HA using analytical flow cytometry. These variants were constructed by the same Golden Gate strategy used above for the library, but by pooling one version of each fragment rather than all versions of each fragment. The assembled plasmid was sequence-verified via Sanger, transformed into the EBY100 yeast strain, plated on

SDCAA-agar, and incubated at 30°C for 48 hr. Single colonies were then restruck onto SDCAA-agar and grown for an additional 48 hr at 30°C for further selection. These restruck colonies were verified to contain the scFv plasmid by colony PCR. Verified colonies were then grown in 5 mL SDCAA with rotation at 30°C for 24 hr; strains were stored by freezing saturated cultures with 5% glycerol at –80°C.

To measure $K_D$, yeast strains were thawed and scFv were induced, incubated with HA antigen, and labeled with fluorophores as described above for the Tite-Seq assay, except yeast cell and antigen volumes were scaled down by a factor of 10. Yeast cell FITC and R-PE fluorescence intensity were then assayed on a BD LSR Fortessa equipped with four lasers (440, 488, 561, and 633 nm), sampling at least 10,000 events per concentration. The equilibrium dissociation constant, $K_D$, were then inferred for each variant $s$ by fitting the logarithm of a Hill function to the mean log R-PE fluorescence for the scFv-expression (FITC-positive) single yeast cells:

$$\text{mean log fluorescence} = \log_{10}\left(A_s \frac{c}{c + K_{D,s}} + B_s\right)$$

where $c$ is the molarity of antigen, $A_s$ is the increase in fluorescence due to saturation with antigen, and $B_s$ is the background fluorescence. All isogenic $K_D$ measurements were made in 2–3 biological replicates (*Figure 1—source data 2*, *Figure 1—figure supplement 3*).

## Fab structural characterization

### Fab production and purification

Antigen binding fragments were cloned and produced in Expi293F cells as above, except the variable heavy chain was cloned into a pVRC expression vector containing the CH1 domain followed by a HRV 3C cleavate site and a 6X His tag. Fabs were purified by cobalt chromatography (Takara) and further purified over an S200 column on an AKTA pure (Cytiva). To the purified Fabs, 1.2 μL HRV 3C protease (Thermo Scientific, #88947) per 200 μg of Fab was added and incubated overnight at 4°C on a roller. The next day, the cleaved Fab was passed over cobalt resin and purified again over an S200 column in 10 mM Tris HCl, 150 mM NaCl, pH 7.5. The resulting Fabs were concentrated to ~15 mg/mL prior to crystallization.

### Fab crystallization

Fabs were crystallized by the hanging drop method. Crystals of unbound UCA Fab with the Y35N (LC) mutation and unbound I-2 Fab with H35N (HC) and Y35N (LC) mutations were grown over solutions of 0.1 M succinic acid (pH 7), 0.1 M bicine (pH 8.5), and 30% polyethylene glycol monomethyl ether 550 or 0.8 M lithium sulfate monohydrate, 0.1 M sodium acetate trihydrate (pH 4), and 4% polyethylene glycol 200 (Hampton Research, #HR2-084), respectively, in a 96-well plate (Greiner, #655101) with ViewDrop II plate seals (sptlabtech, #4150-05600). Crystals were apparent after ~5–7 days. Then, 1 μL of 12% (+/-)-2-methyl-2,4-pentanediol (MPD) in the corresponding solution was added for cryoprotection. The crystals were then harvested and flash-cooled in liquid nitrogen.

### Fab structure determination

X-ray diffraction data was collected at the Advanced Photon Source using beam line 24-ID-E. Diffraction data was processed using XDSGUI (https://strucbio.biologie.uni-konstanz.de/xdswiki/index.php/XDSGUI). Both Fabs reported here were solved by molecular replacement using PHASER in the PHENIX-MR GUI (*Adams et al., 2002*; *McCoy et al., 2007*) by searching with the UCA Fab (PDB: 4HK0) (*Schmidt et al., 2013*) with the HCDR3 deleted and separated into the VH, VL, CH, and CL domains. Refinement was performed in PHENIX (*Adams et al., 2002*) by refining the coordinates and B factors before model building (i.e., the HCDR3) in COOT (*Emsley et al., 2010*). Additional placement of waters and Translation Libration Screw (TLS) refinement followed. The UCA with Y35N Fab showed density for the HCDR3, which was built, but this loop exhibited large B factors (*Figure 4—figure supplement 1*). The I-2 with H35N and Y35N Fab showed no clear density for the HCDR3 or the LCDR2 so these were removed from the structure. The resulting structures were validated using MolProbity (*Chen et al., 2010*) prior to deposition at the Protein Data Bank (8EK6 and 8EKH).

## Antibody-antigen binding kinetics measurements

Kinetics measurements were acquired on an Octet RED96e (Sartorius). To mimic the interaction between yeast-displayed scFv and trimeric HA, IgG was loaded onto Anti-Human Fc Capture (AHC) biosensors (Sartorius, #18-5060). To reduce the avidity effect, IgGs were loaded to a density of ~0.1 nm using a solution of 10 nM of IgG. All binding measurements were obtained in 'kinetics' buffer: PBS supplemented with 0.1% BSA and 0.01% Tween20. Binding measurements were acquired as follows with shaking at 1000 rpm – baseline: 60 s; loading: 30 s with threshold at 0.1 nm; baseline: 60 s; association: 360 s; dissociation: 600 s. Tips were regenerated a maximum of four times by alternating between 10 mM glycine (pH 1.7) and kinetics buffer three times with 10 s in each buffer. Kinetics measurements were obtained at four temperatures for each antibody: 20, 25, 30, and 35°C. Kinetics measurements for the UCA, I-2, and CH65 were also acquired at 40°C. Prior to each measurement, the plate was allowed to equilibrate to the set temperature for 20 min. Each full-length, trimeric HA (MA90, MA90-G189E, and SI06) was assayed at six concentrations: 500, 250, 125, 62.5, 31.25, and 15.625 nM. For each antibody against each HA, antibody assayed with buffer only was used as a reference for subtraction. Additionally, each run contained an irrelevant IgG (CR3022) at the highest HA concentration (500 nM) to detect any nonspecific interaction, which was at background level. To account for the multivalency of the analyte (trimeric HA), the bivalent analyte model was used for global curve fitting in the Sartorius Data Analysis HT software version 12.0.2.59.

## Acknowledgements

We thank Zach Niziolek for assistance with flow cytometry and members of the Desai lab for helpful discussions. We thank Jesse Bloom for providing the plasmids required to generate recombinant influenza viruses and Nicholas Heaton for sharing influenza virus-related protocols. AMP acknowledges support from the Howard Hughes Medical Institute Hanna H Gray Postdoctoral Fellowship, TD acknowledges support from the Human Frontier Science Program Postdoctoral Fellowship, AGS acknowledges support for NIH grants R01AI146779 and P01AI89618-A1, MMD acknowledges support from the NSF-Simons Center for Mathematical and Statistical Analysis of Biology at Harvard University, supported by NSF grant no. DMS-1764269, and the Harvard FAS Quantitative Biology Initiative, grant DEB-1655960 from the NSF and grant GM104239 from the NIH. Computational work was performed on the FASRC Cannon cluster supported by the FAS Division of Science Research Computing Group at Harvard University. We thank the beamline staff at NE-CAT for help with data collection; this work used NE-CAT beamlines (GM124165), a Pilatus detector (RR029205), an Eiger detector (OD021527) at the APS (DE-AC02-06CH11357).

## Additional information

### Competing interests

Angela M Phillips: has recently consulted for Leyden Labs. Michael M Desai: recently consulted for Leyden Labs. The other authors declare that no competing interests exist.

### Funding

| Funder | Grant reference number | Author |
|---|---|---|
| Howard Hughes Medical Institute | Hanna H. Gray Postdoctoral Fellowship | Angela M Phillips |
| Human Frontier Science Program | Postdoctoral Fellowship | Thomas Dupic |
| National Institutes of Health | R01AI146779 | Aaron G Schmidt |
| National Institutes of Health | P01AI89618-A1 | Aaron G Schmidt |
| National Science Foundation | DMS-1764269 | Michael M Desai |

| Funder | Grant reference number | Author |
| --- | --- | --- |
| National Science Foundation | DMS-1655960 | Michael M Desai |
| National Institutes of Health | GM104239 | Michael M Desai |

The funders had no role in study design, data collection and interpretation, or the decision to submit the work for publication.

## Author contributions
Angela M Phillips, Conceptualization, Formal analysis, Supervision, Validation, Investigation, Visualization, Methodology, Writing – original draft, Writing – review and editing; Daniel P Maurer, Conceptualization, Formal analysis, Validation, Investigation, Visualization, Methodology, Writing – original draft, Writing – review and editing; Caelan Brooks, Formal analysis, Validation, Investigation, Visualization, Methodology, Writing – review and editing; Thomas Dupic, Formal analysis, Visualization, Methodology, Writing – review and editing; Aaron G Schmidt, Michael M Desai, Conceptualization, Supervision, Funding acquisition, Writing – review and editing

## Author ORCIDs
Angela M Phillips ⓘ http://orcid.org/0000-0002-9806-7574
Daniel P Maurer ⓘ http://orcid.org/0000-0003-2074-5416
Michael M Desai ⓘ http://orcid.org/0000-0002-9581-1150

## Decision letter and Author response
Decision letter https://doi.org/10.7554/eLife.83628.sa1
Author response https://doi.org/10.7554/eLife.83628.sa2

# Additional files

## Supplementary files
• Supplementary file 1. Plasmid map of pCHA with UCA860 scFv sequence.
• Supplementary file 2. Plasmid map of pCHA with CH65 scFv sequence.
• Supplementary file 3. Primers for constructing CH65 combinatorial mutation library.
• Supplementary file 4. Primers for sequencing library preparation.
• Supplementary file 5. MA90 full-length soluble ectodomain HA expression construct.
• Supplementary file 6. MA90-G189E full-length soluble ectodomain expression construct.
• Supplementary file 7. SI06 full-length soluble ectodomain HA expression construct.
• MDAR checklist

## Data availability
Data and code used for this study are available at https://github.com/amphilli/CH65-comblib, (copy archived at swh:1:rev:cea336dfd05fa31d675f79038428bf7f0d177e78). Antibody affinity and expression data are also available in an interactive data browser at https://ch65-ma90-browser.netlify.app/. FASTQ files from high-throughput sequencing are deposited in the NCBI BioProject database under PRJNA886089. X-ray crystal structures of the Fabs reported here are available at the Protein Data Bank (8EK6 and 8EKH).

The following datasets were generated:

| Author(s) | Year | Dataset title | Dataset URL | Database and Identifier |
| --- | --- | --- | --- | --- |
| Phillips AM, Maurer DP, Brooks C, Dupic T, Schmidt AG, Desai MM | 2022 | Binding affinity landscape of CH65 to divergent influenza H1 strains | https://www.ncbi.nlm.nih.gov/bioproject/PRJNA886089/ | NCBI BioProject, PRJNA886089 |

*Continued on next page*

*Continued*

| Author(s) | Year | Dataset title | Dataset URL | Database and Identifier |
|---|---|---|---|---|
| Phillips AM, Maurer DP, Brooks C, Dupic T, Schmidt AG, Desai MM | 2022 | CH65 Tite-Seq KD Measurements | https://ch65-ma90-browser.netlify.app | CH65 online data browser, ch65-ma90 |

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
