## [Editor Report]

In this valuable study, the authors convincingly show that epistasis between mutations plays a role in the evolution of broadly neutralizing influenza antibodies. The authors provided accurate descriptions within the text and included a graphic summary focusing on the epistatic and non-epistatic models.

---

## [Decision Letter]

**Decision letter after peer review:**

Thank you for submitting your article "Hierarchical sequence-affinity landscapes shape the evolution of breadth in an anti-influenza receptor binding site antibody" for consideration by *eLife*. Your article has been reviewed by 2 peer reviewers, and the evaluation has been overseen by a Reviewing Editor and Betty Diamond as the Senior Editor. The reviewers have opted to remain anonymous.

Essential revisions:

(1) As described in review 1 (recommendation for the authors), some editorial changes are required to make a more accurate description.

(2) As described in review 2 (recommendation for the authors), a graphic summary, focusing on the epistatic and non-epistatic models, is required.

*Reviewer #1 (Recommendations for the authors):*

1. Line 357: The statement on two of the I-2 mutations (G31D and M34I) is not very convincing. As shown by the unconstrained paths in Figure 5B, the mutation order probabilities of G31D and M34I span quite evenly across most mutation orders under MA90, SI06 as well as the optimal scenario.

2. It will help readers to follow the flow of the manuscript if the Results section includes subheadings.

3. Lines 219-220: "(e.g., Y35N, Y48C, D49Y, G31D, Y33H, H35N, N52H)". It is unclear which chain (heavy or light) these mutations belong to. This issue is present throughout the Results section. Is there a way to specify light chain and heavy chain residues? That will help improve readability.

4. Lines 267-268: "… we compared the crystal structures of the unbound UCA, I-2, and CH65 as well as CH65 bound to SI06." It might be a good idea to specify that both UCA and I-2 crystal structures have H35N and Y35N mutations here. It was not mentioned elsewhere until line 963.

5. Lines 283-286: "Thus, while N52H alone confers affinity to MA90-G189E, G31D and M34I (I-2) reduce the dissociation rate by ~3.5-5 fold to improve affinity (Figure 4C, Figure 4—figure supplement 5). Notably, these interactions are distant from the site of viral escape (G189E, Figure 4B, 4C)." However, based on Figures 4B and 4C, it seems like G31D and N52H are quite close to G189E.

6. Line 299: "… Y35F mutation upon affinity maturation which is only a removal of a hydroxyl group". Does Y35F also remove a hydrogen bond between LFWR2 and HCDR3?

7. Is it possible that the different HCDR3 conformations among UCA (PDB 4HK0), I-2 (PDB 4HK3), CH65 (PDB 4WUK), UCA+Y35N, and I2+Y35N+H35N can be in part explained by the difference in crystal packing since these structures have different space groups?

8. In Figure 5A and Figure 5 —figure supplement 1, the meanings of "most likely", "various orders", and "other orders" need to be clarified in the figure legends.

9. The specific I-2 mutations (G31D, M34I, N52H) are not mentioned until line 256. It would be nice to have a sentence specifying them when I-2 is first brought up.

*Reviewer #2 (Recommendations for the authors):*

The landscapes that the authors show can be presented in schematic views as a graphic summary in supportive figures, focusing on the epistatic mode and non-epistatic mode. This might be helpful for readers to locate the position in the antibody at a glance.

---

## [Author Response]

Essential revisions:(1) As described in review 1 (recommendation for the authors), some editorial changes are required to make a more accurate description.

These changes have been made and are addressed point-by-point in response to Reviewer #1.

(2) As described in review 2 (recommendation for the authors), a graphic summary, focusing on the epistatic and non-epistatic models, is required.

We have added a graphic summary focusing on the epistatic and non-epistatic models, and present this as Figure 5—figure supplement 3.

Reviewer #1 (Recommendations for the authors):1. Line 357: The statement on two of the I-2 mutations (G31D and M34I) is not very convincing. As shown by the unconstrained paths in Figure 5B, the mutation order probabilities of G31D and M34I span quite evenly across most mutation orders under MA90, SI06 as well as the optimal scenario.

We agree that the probability of M34I is relatively even up to mutational order ~12. Though G31D is likely to occur up to mutational order ~10, it is amongst the most likely mutations to occur early in maturation (*e.g.,* mutational orders 1-3). We have accordingly modified our description of the data:

“Even when we do not constrain pathways to pass through I-2, we find that two of the I-2 mutations (G31D and N52H) and the epistatic mutations that interact with HCDR3 (e.g., Y33H, H35N, and previously uncharacterized Y35N) are most likely to occur early in mutational trajectories, especially in scenarios that begin with MA90 selection.”

2. It will help readers to follow the flow of the manuscript if the Results section includes subheadings.

We have added subheadings to the manuscript.

3. Lines 219-220: "(e.g., Y35N, Y48C, D49Y, G31D, Y33H, H35N, N52H)". It is unclear which chain (heavy or light) these mutations belong to. This issue is present throughout the Results section. Is there a way to specify light chain and heavy chain residues? That will help improve readability.

We have specified what chain each mutation is located in throughout the Results section using V_H_ and V_L_ notation.

4. Lines 267-268: "… we compared the crystal structures of the unbound UCA, I-2, and CH65 as well as CH65 bound to SI06." It might be a good idea to specify that both UCA and I-2 crystal structures have H35N and Y35N mutations here. It was not mentioned elsewhere until line 963.

We agree that these lines may be confusing. The crystal structures referenced here are that of the UCA, I-2, and CH65 (without the Y35N and/or H35N mutations in the UCA or I-2). In addition to the previously determined structures, we also determined new structures of the UCA and I-2 with the Y35N and/or H35N mutatons. We've added text to clarify the distinction.

5. Lines 283-286: "Thus, while N52H alone confers affinity to MA90-G189E, G31D and M34I (I-2) reduce the dissociation rate by ~3.5-5 fold to improve affinity (Figure 4C, Figure 4—figure supplement 5). Notably, these interactions are distant from the site of viral escape (G189E, Figure 4B, 4C)." However, based on Figures 4B and 4C, it seems like G31D and N52H are quite close to G189E.

We thank the reviewer for bringing this to our attention. Although the distance between the escape mutation (G189E) and the mutations G31D and N52H is quite large (~15-20Å), this is difficult to infer from the figures. We've accordingly added a new panel to Figure 4C to clearly show this distance between these residues and modified the text to reflect this:

"Notably, these interactions are distant (~15-20Å) from the residue conferring viral escape, precluding any direct interaction (Figure 4B, 4C)."

6. Line 299: "… Y35F mutation upon affinity maturation which is only a removal of a hydroxyl group". Does Y35F also remove a hydrogen bond between LFWR2 and HCDR3?

We have clarified the text to reflect that lack of a hydroxyl group in the phenylalanine side chain prohibits hydrogen bonding.

7. Is it possible that the different HCDR3 conformations among UCA (PDB 4HK0), I-2 (PDB 4HK3), CH65 (PDB 4WUK), UCA+Y35N, and I2+Y35N+H35N can be in part explained by the difference in crystal packing since these structures have different space groups?

We thank the reviewer for bringing up this important point. Crystal packing can influence the CDR conformations of unbound Fabs. Indeed, in previously published work, the unbound UCA (4HK0) had two of the copies of the Fab in the asymmetric unit: crystal packing constrained one HCDR3 but the resulted in the other projected into solvent; for the former copy, this resulted in electron density clearly defining the HDCR3, but for the latter it did not. It was inferred that this reflected the inherent disorder of the UCA HCDR3 with many possible conformations. For I-2 (4HK3), crystal packing resulted in the constrained conformation. Here, for UCA+Y35N (8EK6) Fabs have crystal-packing contacts that influence the observed conformation. For I-2+Y35N+H35N Fab, the HCDR3 projects into solvent with no clear electron density suggesting that it can adopt many different conformations. In contrast, the unbound mature antibodies CH65 (4WUK) and CH67 (4HKB) show the same conformation observed in the antigen-bound Fabs: CH65 bound to SI06 HA (5UGY) and CH67 bound to SI06 HA (4HKX). Because the unbound conformations in the non-mature Fabs are influenced by crystal packing (as the reviewer points out), we cannot make any statements about those particular observed conformations, only that the HCDR3s in those Fabs can adopt multiple alternative conformations and are not rigidified as is the case for the affinity-matured antibodies. We've modified text to clarify these points:

"Previous studies on this lineage^4,54^ showed that HCDR3 rigidification contributed to high-affinity binding to SI06: crystal structures of the unbound, affinity-matured Fabs had the same HCDR3 configurations as those in the antigen-bound state; the UCA and I-2, however, were either disordered or constrained, due to crystal packing, in a binding-incompatible state. To determine whether the mutations Y35N and H35N could stabilize a binding-compatible HCDR3 conformation, we determined x-ray crystal structures of unbound Fabs containing Y35N in the UCA background or Y35N and H35N in the I-2 background and compared them to previously determined structures (Figure 4—figure supplement 1C)."

8. In Figure 5A and Figure 5 —figure supplement 1, the meanings of "most likely", "various orders", and "other orders" need to be clarified in the figure legends.

We have clarified this in the respective figure legends.

9. The specific I-2 mutations (G31D, M34I, N52H) are not mentioned until line 256. It would be nice to have a sentence specifying them when I-2 is first brought up.

We have specified the I-2 mutations early in the Results section.

Reviewer #2 (Recommendations for the authors):The landscapes that the authors show can be presented in schematic views as a graphic summary in supportive figures, focusing on the epistatic mode and non-epistatic mode. This might be helpful for readers to locate the position in the antibody at a glance.

We have added a graphic summary of the epistatic and non-epistatic models and the corresponding landscapes as Figure 5—figure supplement 3. We have also included a summary of the structural basis of the epistasis in the graphic summary and have also provided a new figure (Figure 3—figure supplement 5) that plots the additive and epistatic effects of mutations on the crystal structure for each antigen.